# How the Brain Becomes the Mind: Can Thermodynamics Explain the Emergence and Nature of Emotions?

**DOI:** 10.3390/e24101498

**Published:** 2022-10-20

**Authors:** Éva Déli, James F. Peters, Zoltán Kisvárday

**Affiliations:** 1Department of Anatomy, Histology, and Embryology, University of Debrecen, 4032 Debrecen, Hungary; 2Department of Electrical & Computer Engineering, University of Manitoba, Winnipeg, MB R3T 2N2, Canada; 3Department of Mathematics, Adiyaman University, Adiyaman 02040, Turkey; 4ELKH Neuroscience Research Group, University of Debrecen, 4032 Debrecen, Hungary

**Keywords:** brain state, Carnot cycle, emotions, consciousness, resting entropy

## Abstract

The neural systems’ electric activities are fundamental for the phenomenology of consciousness. Sensory perception triggers an information/energy exchange with the environment, but the brain’s recurrent activations maintain a resting state with constant parameters. Therefore, perception forms a closed thermodynamic cycle. In physics, the Carnot engine is an ideal thermodynamic cycle that converts heat from a hot reservoir into work, or inversely, requires work to transfer heat from a low- to a high-temperature reservoir (the reversed Carnot cycle). We analyze the high entropy brain by the endothermic reversed Carnot cycle. Its irreversible activations provide temporal directionality for future orientation. A flexible transfer between neural states inspires openness and creativity. In contrast, the low entropy resting state parallels reversible activations, which impose past focus via repetitive thinking, remorse, and regret. The exothermic Carnot cycle degrades mental energy. Therefore, the brain’s energy/information balance formulates motivation, sensed as position or negative emotions. Our work provides an analytical perspective of positive and negative emotions and spontaneous behavior from the free energy principle. Furthermore, electrical activities, thoughts, and beliefs lend themselves to a temporal organization, an orthogonal condition to physical systems. Here, we suggest that an experimental validation of the thermodynamic origin of emotions might inspire better treatment options for mental diseases.

## 1. Introduction

“The laws of nature are written in the workings of our brains”—Thomas L. Saaty.

Experimental and computational neuroscience aims to better understand the brain from the molecular to the system levels [1,2,3]—but the higher-order functions such as consciousness persistently resisted conventional approaches [4]. Although the scientific understanding of emotions has been challenging, their nature is of utmost importance in psychology, education and AI research.

Life relies on energy, connecting sensory input to consumption at the most basic level. This simple relationship evolved into the unfathomable complexity of the human brain. For example, image representation in a digital camera is a series of 1′s and 0′s on a memory card, but it is impossible to uncover such causal relationships in perception. Instead, the brain’s equilibrium processes show analogies to the overarching principles of the physical world. Feelings and rumors spread like heat in liquids. Sentiment contagion is the social spreading of sentiments according to a decay function [5,6], based on credibility, self-assertiveness, and other qualities. The physical sciences have provided a dramatic new set of mathematical and physical tools for studying intelligence and agency [7,8].

The dramatically unique qualities of the mind have produced significant debates. Intimately linked with the self, the mind is considered qualia in philosophy [9]. Mental contents, such as ideas, beliefs, and emotions, have temporal existence without familiar physical qualities. Connecting the physiological manifestations of consciousness to the brain’s electric activities might explain how the brain gives rise to a rich inner life, the introspectively accessible and ephemeral inner cosmos, which is at the heart of the mind-body problem.

How much information do we gain when we learn something? The surprising answer might be that learning introduces potential (in the form of complexity and organization), which determines behavior and response. Mental energy is the confidence to set meaningful goals and the steady, optimistic focus to reach for them. The essence of learning might be the counterintuitive process of increasing the degrees of freedom via information erasing. Therefore, Landauer’s principle might explain intelligent computation.

For example, the anatomical layout and the synaptic weights of recurrent synaptic connections store memories [10]. Therefore, spontaneous activities reflect the priors residing in the network architecture, which is susceptible to experience and learning-dependent modifications throughout life. Such an intelligent computation can be viewed as imperfect copying and selecting neural informational patterns according to the Darwinian process. It is thus conjectured that the Bayesian update and replicator dynamics of the above process is an ideal operation for record-keeping optimization and shaping activations [11,12,13].

Intelligent responses, in general, depend on a precise mental model [10]. In other words, we find that “every good regulator of a system must be a model of that system” [14]. The sensory system projects the physical environment into the brain by an active, energy-requiring process often independent of conscious intention. For example, environmental noise, disturbing light, or noxious fumes can intrude on current activity and even sleep [15]. Thus, the brain adapts to its environment by adopting the laws of physics.

A fundamental feature of these cognitive phenomena is temporal dependence [16,17], which represents an orthogonal projection [18]. As material systems observe the principle of least action when moving in space, biological systems optimize their action repertoire between the past and the future. For example, adjusting muscle strength throughout movement execution requires anticipating and coordinating a precise sequence. Recent findings showing the irreversibility of resting brain activations permitted a meaningful comparison of entropy with physical systems. In short, we may consider the mind’s predictive processing as the temporal equivalent of stationary action in physics. Consciousness, often synonymous with the mind, is adaptive, self-regulating, and homeostatic [19]. Similarly to a gravitational imbalance, threats to personal security or ego trigger conscious protective measures. For example, post-traumatic stress disorder (PSTD) keeps a total grip on the mind. Shame, particularly chronic shame, produces high frequency and mentally taxing mind-wandering [20]. Therefore, transgressions can cause remorse, shame, and regret, dominating random thoughts.

In sum, emotions or mental states represent internal drives based on a deviation of vital bodily parameters from equilibrium [21]. Therefore, emotions, the ultimate source of actions, are context-driven motivations, the fundamental forces of the mind. In this view, free will is a graded ability eroded by exothermic processes, such as depression and anxiety. The following discussion will deduce emotions and the thermodynamic consequences of awareness. In this, we follow earlier insightful work on the free energy principle and the thermodynamic analysis of cognition [22,23,24,25,26].

In the Section 1, we describe the temporal brain, the Section 2 discusses entropy, and Section 3 focuses on the thermodynamic analysis of the neural system with particular emphasis on the thermodynamics of emotions. The Section 4 focuses on spontaneous processes. We end with conclusions.

## 2. The Temporal Mind

Consciousness is part of the physical environment [27,28,29] and must be physical and explained by physical principles [27,30]. Let us start with perception, a highly involuntary [31] sensory influx [32,33], and comprehension [34]. The synchronization of millions of neurons and functional interaction between specific microstates [35] give rise to characteristic and measurable frequencies. Recent analyses of brain activities have proven that electric activities correlate with the neuropsychology of consciousness. The brain waves report on behavior, emotions, and thoughts.

(1)Frequency dependence

The brain waves have been grouped according to their frequencies. Higher frequencies have higher energy needs and larger information-carrying capacity, leading to a power-law relationship of electromagnetic activities [36]. Wave-like cortical fluctuations can anticipate events and the consequences of actions [37,38], such as the global coordination of muscle tone by the evoked potential in motor cortices [39]. The information flow of synaptic connections represents momentum, causing thoughts to be spontaneous, unexpected, and challenging to control or retrace [40,41]. These activations interact with the brain’s internal oscillation pattern, a highly fluid and malleable mental background, turning sensory perception subjective.

(2)Temporal organization

Identification with the body is the basis of homeostatic self-regulation [42]. The highly autonomous regulation, an apparent consciousness requirement, supports a temporal integration. The biological dependence on air, water, rest, and food also leads to a temporal organization. In contrast to the tangible presence of physical objects, mental concepts, such as ideas, beliefs, and emotions, have temporal existence without weight or other physical qualities.

Self-sense is an emergent personal perspective connected to the brain’s resting state [43,44]. As the embodiment of self, it ensures temporal continuity throughout life and enables the attribution of mental states, beliefs, intents, desires, emotions, and knowledge to other beings and even objects. Discrete order associations in learning, speech, muscle coordination [44,45], and neural processing [46,47] exist on many time scales. The temporally nested and discrete mental constructs may include the individual’s retrospective lifetime (Figure 1).

Therefore, cognition is built of substantial periods of continuous unconscious processing followed by discrete conscious percepts and believes [48]. The alternation between unconscious versus conscious, continuous versus discrete, fluid charge flow versus thermodynamic balance represents cognition’s dual nature, which explains some of the hard to define qualities of the mind. The above differences might be reminiscent of the classical and quantum divide in quantum mechanics.

(3)Associative model

The integration of separate events generates an associative model. Disjointed sensory clues coalesce into coherent representations of a first-person, unified perspective [42,49,50,51]. Intelligent systems’ fundamental unity [27] permits predictive processing [4], the optimization of action repertoire between the past and the future [8,52], and between memory and expectation. The stationary-action principle states that the action for actual trajectories is a minimum. Likewise, predictive processing charts an optimal or least temporal action [53], which mitigates the utter reliance on the environment. Therefore, controlling the future might be a primary objective of cognitive function (Table 1). Robotic studies confirm that future orientation is an essential requirement of intellect [54].

## 3. Considerations of Entropy

### 3.1. Similarities of the Brain and Physical Systems Based on Entropy

Entropy measures the variety of existing possible configurations within a system. Since entropy production directs all changes toward equilibrium, it is connected to time’s arrow [55,56,57]. Therefore, time can be defined as the entropy generation rate, often associated with thermal disorder and loss of work capacity [58]. Similarly, entropy production is a fundamental requirement of the brain’s normal self-organizing energy functioning.

Cortical wave-like fluctuations, the so-called traveling waves [40,41], are the fundamental elements of cortical information processing [59]. It is also known that the activity patterns generated by cortico-cortical circuits actively shape the evoked cycle by modulating neural and perceptual sensitivity [60]. The neural system’s momentum trajectory reflects forces acting on different time scales. Initially, the activations are low-dimensional. Entropic effects slow the flows on the task-specific activations serving as a principal enabling a link between neural activity and behavior. The evoked cycle’s state vector represents the directionality and size of attitude [8,61,62,63,64,65,66,67,68].

The entropy-generating resting activations show apparent irreversibility [69,70]. However, brain entropy correlates with the number of *accessible* neural states (degrees of freedom) or surprise potential (Table 1). Rather than loss of work potential, it represents increasing degrees of freedom, novelty, creativity [71], and intellectual capacity [72] in verbal and performance measures [72,73]. The surprise potential or signal variability is a critical feature of brain function [74].

### 3.2. Entropic Differences between the Brain and Physical Systems

There are notable differences between physical systems and the brain. For example, information input generates a memory, changing the brain (Figure 2). Furthermore, the resting state is a relaxation, which equilibrates the system in preparation for the new cycle. The brain’s temporal rhythms form discrete processing centering on the resting state [7,8], forming a closed thermodynamic cycle.

In physical systems, the free energy initiates work, but the equilibrium position lacks work potential. In contrast, motivational thought initiates work in the resting brain. Likewise, low entropy physical systems move toward equilibrium via irreversible activations. However, reversible activations stabilize the energy-enriched brain, turning low entropy into a stable, although precarious position, which can degenerate into pathologic conditions, depression, mania, psychosis, and schizophrenia [75,76].

Resting entropy measures the thermodynamic free energy and thus regulates stimulus response and spontaneous behavior. Only the low entropy brain has free energy for aggravation and criticism. The above differences from physical systems have challenged the intuitive understanding of neuronal system dynamics.

## 4. Thermodynamic Regulation of the Neural System

What is an intelligent answer to a stimulus? Intuition is based on the physical world’s organizational and operational principles [53]. Spontaneous resting activations, which constantly realign with the external world and consequences of action, are at the center of brain processing [43,44,47]. Thus, the sense of self [44,75,77] is an outgrowth of an internal world model.

The resting state is the basis of the brain and, consequently, the body’s homeostatic regulation. The recurring constancy of the resting state formulates the sense of self, establishing it as the ground state of cognition [61]. During energy-information exchange with the outside environment, the sensory stimulus passes through successive regulatory layers to the associate areas, forming a closed cycle [7,29,43,78,79,80].

Therefore, the evoked activations form a thermodynamic cycle. Accordingly:A stimulus triggers activation, which represents momentum and direction.Entropy-maximizing influences continuously adjust the large-scale spatial synchronization of oscillatory activity.The relaxation that recovers the resting state prepares the system for reactivation.Every cycle changes the brain’s synaptic balance and organization.

The thermodynamic insight of the neural system is an increasingly pursued goal in consciousness science [7,8,23,24,26,80]. The brain’s intelligent computation is based on the sensory system’s information gathering. Sensory stimuli project information about the environment to the cortex via frequency coding. The overall energy availability in the brain is thought to be constant but can display substantial local differences, which can be used to measure energy utilization during stressful and positive conditions by simultaneous EEG-fMRI analysis. Other studies might include broadband near-infrared spectroscopy, which can measure energy metabolism in brain cells’ mitochondria [81].

Accordingly, brain activation and synaptic changes are proportional to the subjective information value of incoming input. For example, the written sign “danger” is perceived differently by toddlers and adults, indicating the observer’s role in the “information” value of a signal [82,83]. Thus, information density determines meaning, preparedness, and context function [84]. Since sensory information’s computational costs depend on the oscillations’ energy requirement, stimuli incorporate the brain into the environment’s energy cycle.

The thermodynamic cycle is reversible. Exothermic processes generate energy, but endothermic processes require energy input. Moreover, the reciprocal accessibility of functional brain states [85] can produce either endothermic or exothermic loops [8,25,28]. Therefore, endothermic cycles absorb energy from the environment, whereas the exothermic brain loses energy.

There is a computational equivalence of operational noise levels (information), therefore arousal [13], with temperature [26]. For example, irritability is accompanied by increased breath, heart rate, skin conductance, shivers, and hot or cold chills, which exert a measurable metabolic burden [86]. Moreover, aggravation, anger, and shame can trigger physical actions. Shouting, pacing, aggression, and other risky behaviors contribute to negative emotions’ high energy costs [87].

Therefore, nervous states, which instigate competition and struggle between community members, represent high social temperatures. Since high-temperature systems are less stable and more sensitive to change, high social temperature represents social volatility.

### 4.1. The Thermodynamics of Emotions

It is increasingly clear that emotions are an integral part of the general neural architecture of the brain [88]. Although it is difficult to distinguish particular emotions, the difference between happy and negative emotions (such as fear, sadness, anger, and disgust) is distinct. Frequency-based binary emotion classification (positive and negative) can achieve 96.81% accuracy [89]. Therefore, based on brain activation profiles, emotional valence is positive during lower frequencies, and negative amid information-heavy higher frequencies [89,90,91].

The brain’s energy/information flow is also frequency-dependent [92,93]. Recent emotion studies underline the cumulative effects of negative emotion (i.e., fear, sadness, anger, and disgust) in the frontal lobe, which is entirely missing for neutral and happy sentiments [89] (Figure 3). For example, remorse and guilt invariably follow mistakes. Moreover, it takes persistent mental effort to alleviate the cognitive burden of these consuming feelings [94,95,96]. In contrast, confidence and satisfaction inspire pleasure and suitable allocation of attentional resources [86].

The above results confirm emotional dependence on meaning-making and psychologically relevant content. Based on the above findings and the established valence-arousal graph used in the literature [89,97], we created a thermodynamic emotion evaluation graph (Figure 4). In a two-scale valence-arousal graph, the vertical axis represents arousal, with zero representing equilibrium and values increasing toward positive (upper half) and negative (lower half) arousal. The horizontal axis represents the ability for mental action, with a lack of mental capacities, such as apathy and depression on the left, with increasing confidence and mental power on the right. The representation also explains the close correlations between similar emotions [89].

#### Emotions as the Fundamental Forces of Motivation

Another point to consider is how the evoked cycle’s thermodynamics determine motivation. Rapid switching of a high-dimensional resting state shrinks the variance as activations collapse into lower-dimensional attractor dynamics [10,98,99] based on whether experiences represent positive or negative charge [100]. The brain devotes an essential thalamic mechanism for distinguishing the direction of emotional signals, relying on the production and release of neurotensin for rewards, but punishment learning comes for free [101]. The above results strongly suggest that positive and negative emotions have contrasting energy metabolisms, and correlate to endothermic or exothermic cycles, respectively.

Metastable, low-dimensional substrates reduce temporal dimensionality [102,103,104] and information density [105,106], focusing attention toward self-preservation. For example, the amygdala is one of the critical structures regulating personal space [107,108]. Violations trigger defensive and alert postures, particularly compelling during high-stakes situations, moral dilemmas, and regrets. Conversely, relaxation depends on reduced amygdala activity [109]. The brain’s memory storage embeds motivation, a state vector, within a multidimensional state or information space [26,88], generating a realistic emotional cinema. The cycle’s energetic consequences arise from the brain’s spatio-temporal orientation [43,46]. Notably, rather than changing the velocity of an object or the speed of the steam engine, it distorts the subjective experience of the moment [110], turning time perception into the force of motivation.

What is the secret of emotions’ action-producing power? Although both positive and negative emotions (low and high frequencies, respectively) slow time perception, they generate opposite energy transformations and motivations [89]. In the case of positive emotions, slow time perception permits relaxation, but high information density causes impatience. In addition, the spectral density of focus highlights cognitive stress, which narrows attention [89,111].

### 4.2. The Endothermic Cycle

Shannon refers to communication systems as the first kind of system, but intelligent systems [54], which rely on temporal computation [4], can control the future. Wissner-Gross and Freer [54] have shown a physical connection between adaptive behavior and entropy maximization. We applied the reversed Carnot (endothermic) cycle to characterize an ideal energy/information cycle with vanishing net entropy production between two cognitive states. To this end, we consider a Bayesian process that can capture a wide range of learning behavior [11,112].

What amount of learning is required from the current knowledge *p* to reach knowledge *q*? When the distribution does not change with the change in the model parameters, the cost function is the Kullback-Leibler divergence ($D_{KL}$), a type of statistical distance and represented as a surprise between the current output and the expected output [12,13].

In the endothermic brain, the discrete $D_{KL}$ is defined as follows:

*X* = discrete random variable in brain signal space *X.*

*p*(*x*) ≥ 0, *q*(*x*) > 0 = probability distribution of *x*, where

*p*(*x*) = probability distribution of observed brain signal *x*,

*q*(*x*) = probability distribution of estimated brain signal *x*.
(1)$$D_{KL}\left((p\left(x\right)\;\|\;q\left(x\right)\right)=\sum_{x\in X}p\left(x\right)\ln\frac{p\left(x\right)}{q\left(x\right)}\geq 0$$
where $\ln\frac{p\left(x\right)}{q\left(x\right)}$ is the novelty of the information. Learning depends on $D_{KL}$ ≥ 0 and when *q* = *p* no learning occurs.

Instead of asking the amount of learning taking place between p and q, we can look for the cognitive updating caused by a stimulus, *F* (the free energy). *S* is the entropy, and *T* is the social temperature as defined by frequencies, *dS* = $\frac{\delta Q}{T}$, therefore
*F* = <*F*> − *TS*(2)
where <*F*> is the expected meaning (intellect). High social temperature increases the stimulus surprise potential; its ability for arousal. The system’s free energy is proportional to the sensory information’s surprise value relative to expectation. As the system moves toward equilibrium, the free energy is minimized. In the endothermic case, the resting entropy increases during stimulus processing, absorbing free energy.

The Boltzmann distribution expresses the probability that a system will be in a state $p_{i}$ as a function of the intellect (energy) and the social temperature, showing an inverse relationship between aggravation (*T*) and intelligent behavior as follows.
(3)$$p_{i}\propto \exp\;\left(-\frac{kF_{i}}{T}\right)$$
where *k* is the Boltzmann constant.

The system evolves through a Markov process.
(4)$$F(q)-F(p)=kT\;D_{KL}$$
where *F*(*q*) is the free energy for *q* and *F*(*p*) is the free energy in *p*.

The system’s free energy is proportional to the sensory information’s surprise value relative to expectation. From Equation (2), we can express the energy requirement of learning
(5)$$\frac{dp_{i}}{dt}=\left(F_{i}-<F>\right)\;p_{i}$$
(6)$$\frac{d}{dt}D_{KL}=-\sum_{i}\left(F_{i}-<F>\right)q_{i}$$
where $\sum_{i}\left(F_{i}-<F>\right)p_{i}$ is the average “relevance” of an incoming stimulus, $D_{KL}$ is how much information is left to learn by going from *p* to *q*. Equation (6) shows the relevance compared to the equilibrium position $p_{i}$. Information with greater surprise changes the brain more significantly. Language, reading, mathematics, and the arts have exponentially increased the information density humans can access. It supports the intuitive notion that we never learn if we are exposed to the same information day in and day out [113,114].

Free energy originates in entropy production by the environment’s exothermic processes, representing the time-reversal asymmetry of time’s arrow [55,56,57]. Surprisingly, time-reversal asymmetry also characterizes the endothermic resting brain (Table 1). In contrast to the physical environment’s increasing disorder, in the brain high resting entropy ensures variability and flexibility [72]. Recent work confirms the energy requiring cell growth in the hippocampus promotes depression resilience [115].

The transience of happy sentiments [89] (Figure 3) does not impose a cognitive burden. The fleeting nature and irreversibility permit future orientation, generosity, intellect [116,117,118], and novelty [69,70], represented by a movement toward the right in Figure 4. Therefore, learning is a continuous increase in confidence that provides energy for emotions, whether happiness or anger.

Landauer’s principle explains learning as the counterintuitive process of increasing the degrees of freedom via information erasing. Learning introduces potential (in the form of complexity and organization), part of memory. Once stored, information is no longer susceptible to noise. Thus, information storage is an energy requiring process [115], analogous to a “phase transition” due to a temperature decrease in a physical system [22]. We have shown that positive memory formation requires energy [115,119] arising from the polar emotional experience processing [101]. Thus, the system can acquire new information in the following cycle while leaving the stored information accessible for decision-making.

In summary, neurotensin production, synaptic reorganization and neurogenesis contribute to the endothermic cycle’s energy need [101,115]. The endothermic cycle confirms the physical relationship between intelligence and entropy maximization [54].

### 4.3. The Exothermic Cycle

Energy generation by exothermic processes is the basis of the second law of thermodynamics. The exothermic cycle in sensory processing might have emerged to correct mistakes during action preparation and execution. Nevertheless, a Bayesian process can turn the exothermic cycle into a mental shift toward anxiety [120], apathy, and depression [76,121], as shown in Figure 4.

The energy state and the entropy of the exothermic brain are smaller than the endothermic condition. Equation (2) shows that high social temperature can turn the free energy negative. Therefore, the free energy changes sign in Equations (1) and (2), because the low entropy state cannot absorb the energy of the stimulus. Although the exothermic state readily increases social temperature, aggravation disperses energy. In addition, anxiety fuels fear conditioning, a stronger, less flexible connectivity. Extensive psychological studies on stress and negative emotions support the above conclusion. Even though the high mental energy brain tends to form an endothermic cycle, high social temperature can produce an exothermic cycle. Thus, a calm mind is most capable of learning and mental advancement.

The exothermic equilibrium is not a high entropy state; the brain responds to even minor difficulty by increasing social temperature (excitement or aggravation): *TS* >> <*F*>, which dramatically changes the cycle’s dynamics. Since the high entropy state occurs early in the cycle, without time for contemplation, the high social temperature fuels energy loss by urgent, thoughtless actions. According to the thermodynamic principles, *Efficiency* = $\frac{W\;}{QH}$ = $\frac{TH-TC}{TH}$ where TC  is the social temperature in the resting state, TH is the social temperature during the first phase of the cycle.

Therefore, the exothermic brain converts heat (*QH*) to work (*W*): *W* = $\frac{TC\;}{TH}$ *QH*.

How can information accumulation lead to adverse psychological symptoms?

Like all exothermic processes [55,56,57], the low entropy brain loses energy (Table 1). For example, it dissipates energy to the environment via criticism, destructive behavior, or violence [7,8].However, in contrast to physical processes, the reversible brain activations stabilize low entropy, creating long-term adverse emotional and psychological outcomes.Long-term potentiation reduces the degrees of freedom due to the loss of synaptic complexity, producing repetitious, monotone thinking [122]. The negative thought pattern becomes more powerful and pessimistic through a Bayesian process, affecting behavior. For example, the severity of cognitive impairment in depression correlates with brain entropy reduction [122].Attention to the past degrades optimism for the future [123,124]. In addition, the energy imbalance may misalign hormone regulation with adverse mental and immune consequences, anxiety, and depression [76,121].

What is the currently available evidence for the above processes? First, negative emotional states have more significant energy requirements than positive emotional or neutral mental states [125], such as gamma rhythms in the prefrontal lobe [89]. In addition, negative emotions (e.g., fear, sadness, anger, and disgust) have cumulative effects [89], which exhaust motivation [126]. Furthermore, the cumulative potential can lead to emotional suppression’s explosive ‘boomerang’ effect [127]. Furthermore, impulsivity enhances the propensity for alcohol, drug, gambling problems, overspending, and overeating [128].

The psychological pressure of information flow causes a sense of time shortage, enhancing time perception [129], which denies a balanced response. The resulting deterministic behaviors [130], such as rumination and repetitive negative thinking [88], represent insecurity, which ultimately may culminate in depression [121], immune problems, mental diseases [131], and adverse health outcomes [132].

In contrast to love, learning, and acceptance, negative emotions project energy out to the environment (through criticism and aggravation). Therefore, analyzing neurotransmitter action and the related energy changes due to cognitive loads might be fruitful. Taken together, the energetic study of mental states may explain their long-term cognitive and health effects [133,134] the understanding of arousal potential [135].

## 5. Spontaneous Processes

Most cognitive and behavioral actions represent a response to a stimulus, but spontaneous actions, although often precipitated by a related stimulus, show no relationship to the stimulus nature or magnitude. For example, a gunman’s shooting spree can be triggered by a job loss, but it results from long-term insecurity and hopelessness.

In thermodynamics, a spontaneous process transpires without external input to the system (Table 2) which can be assessed by the Gibbs free energy equation.
ΔG = ΔH − ΔTS (7)
where G is the Gibbs free energy and H is enthalpy ≈ internal or mental energy, the function of intellect, creativity, and mental flexibility.

In exothermic changes ΔH < 0 or ΔTS > 0. However, endothermic process (ΔH > 0) can occur in supportive environments.

The free energy principle and active inference can provide a mathematical argument for spontaneous processes, including emotional ones. The time-evolution of a system moves to thermodynamically more stable energy state (closer to thermodynamic equilibrium). The organism releases free energy in an exothermic process, which increases the entropy of the environment (e.g., criticism, anger, and brutality). Inversely, an endothermic process absorbs free energy from the environment. Therefore, emotions are part of a thermodynamic regulation of cognition, a fundamental part of the evolution of intellect.

Commonly, spontaneous behavior is called “free will” since it feels natural and easy. The exothermic cycle leads to irrational, damaging behavior, which, in extreme cases can lead to depression, disease, or the individual’s demise.

One may conceive that hemispheric asymmetry could result in uneven thermodynamics between the brain hemispheres [136] but the structural analysis of intra- and inter-lobe connectivity is beyond the present work.

## 6. Conclusions

Neuroscience increasingly produces new developments in psychology and social sciences by using physical principles. The sensory system records the physical world’s spatial organization via temporal rhythms to form discrete processing centered on the resting state. Therefore, perception forms a closed thermodynamic cycle and can be analyzed via the reversible Carnot cycle.

We used the free energy principle to analyze emotions and spontaneous behavior. We show that resting entropy is a measure of the thermodynamic free energy, regulating stimulus response and spontaneous behavior. Since stimulus supplies energy for endothermic processes, healthy mental functioning depends on the reversed Carnot cycle. Love, learning, and acceptance accumulate mental energy, fueling confidence, creativity, and future orientation. Indeed, high resting entropy permits synaptic flexibility and complex behavior.

In contrast, the Carnot cycle radiates energy out via an exothermic process. Although the evolutionary purpose of negative emotions might have been correcting mistakes, the Bayesian reinforcement of heat production expels heat via criticism and aggression, engendering insecurity and regret. Moreover, passive (non-action forming) information accumulation may compromise hormonal regulation.

The thermodynamic analysis of the brain’s evoked cycle shows that emotions have action-producing power, forming the fundamental forces of motivation. Furthermore, the duality represented by the brain continuous information processing and the quantized nature of intellectual changes and believes might be reminiscent of quantum and classical discrepancy. Therefore, biological organisms adapt to their environment by adopting the laws of physics. Generalizing the grand theories of physics to temporal systems might inspire a better understanding of cognition.

The brain devotes an essential thalamic mechanism for distinguishing the direction of emotional signals, relying on the production and release of neurotensin for rewards, but punishment learning comes for free [101]. Publications by reputable laboratories support our working hypothesis on the contrasting metabolisms positive and negative emotions. Measuring the brain energy utilization during stressful and positive conditions by simultaneous EEG-fMRI analysis and measuring the energy metabolism in brain cells’ mitochondria by broadband near-infrared spectroscopy can verify our working hypothesis. 

We are optimistic that the results can inspire novel treatment options for mental disease and other problems.

## Figures and Tables

**Figure 1 entropy-24-01498-f001:**
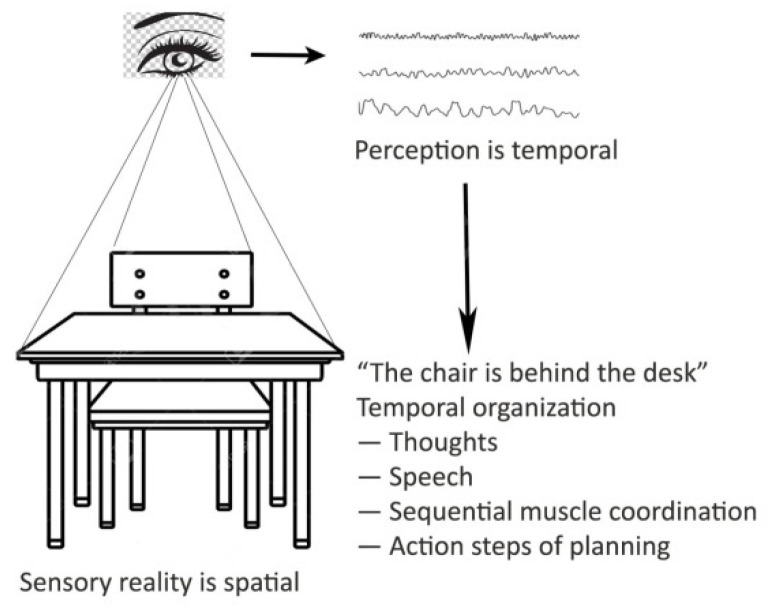
The orthogonality of perception is a temporal organization. The chair behind the desk is a spatial organization, represented by brain frequencies and expressed by a temporal order of speech, thoughts, and actions.

**Figure 2 entropy-24-01498-f002:**
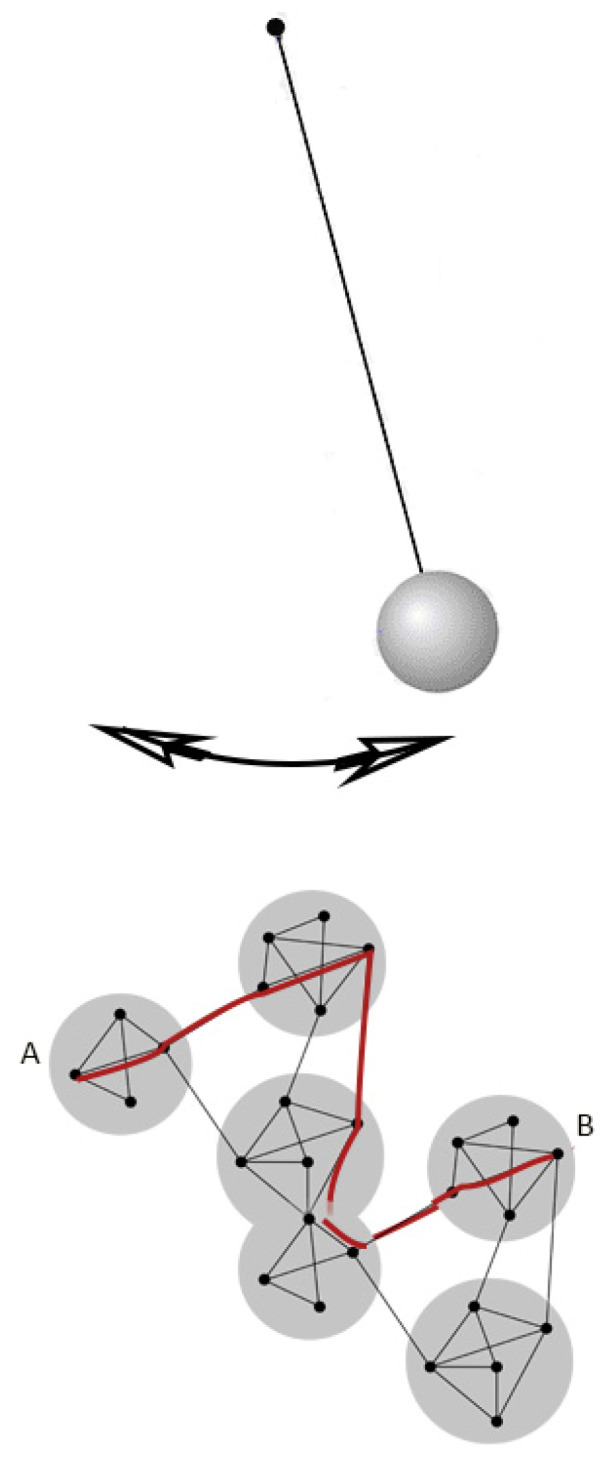
The analysis of neural activation as a harmonic motion. The system returns to its starting position without creating memory (**top**). In a symbolic representation of the neuronal activation, red color represents a path between the brain’s modular structures (**bottom**). The vector path (A–B) is determined by the system’s memory and the stimulus’s qualities.

**Figure 3 entropy-24-01498-f003:**
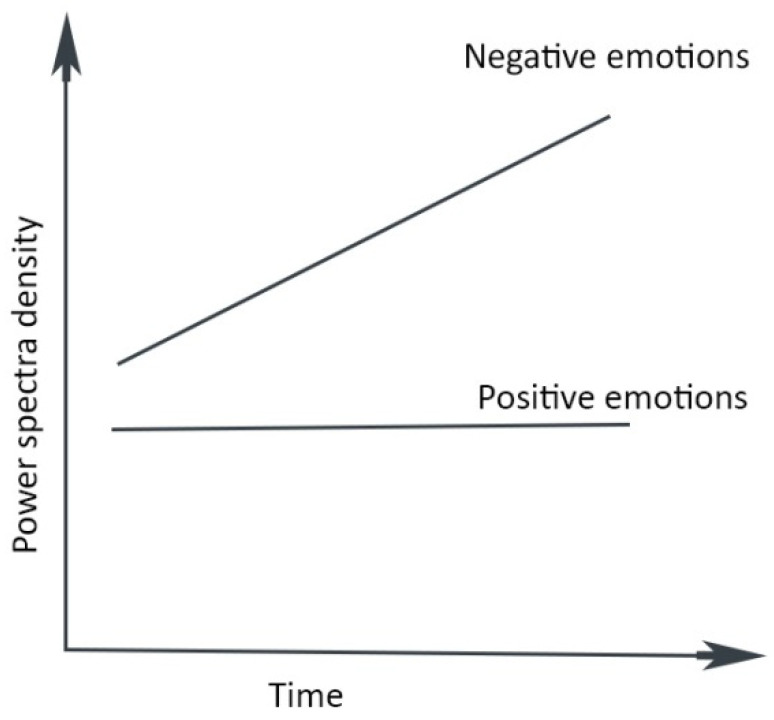
The cognitive burden of positive and negative emotions (drawn following [89]).

**Figure 4 entropy-24-01498-f004:**
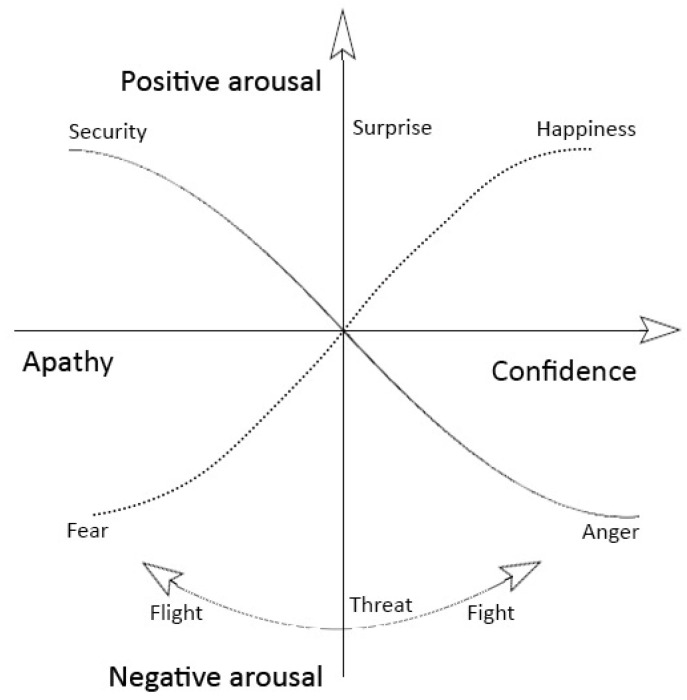
The characterization of emotions based on energy. The horizontal axis represents the mental abilities (mental energy), which change from an inability, such as apathy and depression, to confidence in mental action on the right. The vertical axis shows the environmental influence on arousal with a neutral state at the origin. Positive arousal generates surprise (top), and threat produces anxiety (bottom). A threat can trigger a fight or flight response.

**Table 1 entropy-24-01498-t001:** A systematic comparison of the thermodynamics of physical and neural systems. The top three rows report on the orientation, evolution, and directionality of the system’s microstates. The lowest four rows report on the consequences of the above differences in the microstates.

	Physical Systems	Brain Activations
Microstates orientation	Oriented in space	Information entropy oriented in time
System evolution	Brownian motion	Wave-like activations founded on memories
Entropic force	Irreversible macroscopic behavior	Irreversible activations
The consequences of irreversibility	The arrow of time	Future orientation, novelty, curiosity, and creativity
High entropy state	Equilibrium	Equilibrium
Consequences of high entropy	Loss of work potential	Intellect, confidence, and a can-do attitude
Energy input lowers the entropy	The system moves away from equilibrium, but irreversibility remains!	Reversible and repetitive activations
Consequences of low entropy	Increasing work potential	Uncertainty, lack of control, and psychological problems

**Table 2 entropy-24-01498-t002:** Analysis of behavior as a function of environmental and personal factors.

	Exothermic Reaction (Mental Energy Loss)	Endothermic Reaction (Mental Energy Gain)
High entropy environment (stress)	Spontaneous behavior	Spontaneous on low social temperature, which permits overcoming the negativity
Low entropy, supportive environment	Spontaneous on high social temperature because the aggravation overcomes the support	Spontaneous behavior

## Data Availability

Not applicable.

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
