# Peer review of "How the Brain Becomes the Mind: Can Thermodynamics Explain the Emergence and Nature of Emotions?"

_entropy, 2022, doi:10.3390/e24101498_

Round 1

Reviewer 1 Report (New Reviewer)

The manuscript "How the brain becomes the mind: can the carnot cycle explain the emergence and nature of emotions?" devoted to quite an interesting consideration of brain activity from the point of view of thermodynamics. The connection of emotions and the Carnot cycle is considered. Unfortunately, this article is in the nature of general philosophical reasoning. The proposed approaches are difficult to evaluate due to the lack of a reliable quantitative assessment. In general, it can be assumed that such research may be published in another journal.

 Separately, the following remarks can be noted:

1) 144 link - the names of the authors are not indicated

2) Links are not in the order of mention. For example: lines 209, 215, 224, 287, 404

3) Figure 1 is of poor quality. There is no link to figure 1 in the text of the article

4) It is not clear why exactly such geometric figures are chosen in Figure 3.

Author Response

We want to thank the reviewer for the input on the manuscript: "HOW THE BRAIN BECOMES THE MIND: CAN THE CARNOT CYCLE EXPLAIN THE EMERGENCE AND NATURE OF EMOTIONS?" Thank you for giving us the opportunity to submit a revised draft of our manuscript. Please find our point by point response to the feedback in red and the revised text in italics below.

The manuscript "How the brain becomes the mind: can the carnot cycle explain the emergence and nature of emotions?" devoted to quite an interesting consideration of brain activity from the point of view of thermodynamics. The connection of emotions and the Carnot cycle is considered. Unfortunately, this article is in the nature of general philosophical reasoning. The proposed approaches are difficult to evaluate due to the lack of a reliable quantitative assessment. In general, it can be assumed that such research may be published in another journal.

Separately, the following remarks can be noted:

We hope our argument will show that our work moves beyond “general philosophical reasoning.” The main argument of our response is summarized in three points:

1) Can the Carnot cycle explain the emergence and nature of emotions? devoted to quite an interesting consideration of brain activity from the point of view of thermodynamics.

Human emotions are very complex and their objective identification is challenging. We show 3.1., On top of P7. 259-264:

Although it is difficult to distinguish particular emotions, the difference between happy and negative emotions (such as fear, sadness, anger, and disgust) is pronounced. Frequency-based binary emotion classification (positive and negative) can achieve 96.81% accuracy [87]. Therefore, based on brain activation profiles, emotional valence tends toward positive during lower frequencies, and negative amid information-heavy higher frequencies [87, 88, 89, 90].

2) The connection of emotions and the Carnot cycle is considered. Unfortunately, this article is in the nature of general philosophical reasoning.

We would like to reason that the implication of the Carnot-cycle for tackling the thermodynamics of emotion has been validated (Deli et al., 2018, 2021; Deli and Kisvarday 2020). Therefore, we have added the following text in page 7, 12, 2, 11:

Second paragraph on P7. 265-274:

Recent emotion studies underline the cumulative effects of negative emotion (i.e., fear, sadness, anger, and disgust) in the frontal lobe, which is entirely missing for neutral and happy sentiments [87] (Figure 3). For example, remorse and guilt invariably follow mistakes. Moreover, it takes persistent mental effort to alleviate the cognitive burden of these consuming feelings [93, 94, 95]. In contrast, confidence and satisfaction inspire pleasure and a suitable allocation of attentional resources [84]. The above results confirm emotional dependence on meaning-making and psychologically relevant content. Based on the above findings and the established valence-arousal graph used in the literature [87, 96], we created a thermodynamic emotion evaluation graph (Figure 4).

To our best knowledge, for the first time, we were able to provide a full analysis of both positive and negative emotions from the free energy principle. As summarized in the Conclusions on P12. 501-503:

We used the free energy principle to analyze emotions and spontaneous behavior. We show that resting entropy is a measure of the thermodynamic free energy, regulating stimulus response and spontaneous behavior.

We considered the mind's predictive processing vis a vis the stationary action in physics, on P2 paragraph 5. Introduction 82-85:

Recent findings showing the irreversibility of resting brain activations permitted a meaningful comparison of entropy with physical systems. In short, we may consider the mind's predictive processing as the temporal equivalent of stationary action in physics.

We inserted a new section on spontaneous processes with a new table showing the thermodynamic consequences of behavior as a function of environmental and personal factors as follows, Section 4, P 11-12. 463-490:

4 Spontaneous processes

Most cognitive and behavioral actions represent a response to a stimulus, but spontaneous actions, although often precipitated by a related stimulus, show no relationship to its nature or magnitude. For example, a gunman’s shooting spree can be triggered by a job loss, but it results from long-term insecurity and hopelessness.

In thermodynamics, a spontaneous process transpires without external input to the system (Table II) which can be assessed by the Gibbs free energy equation. 

ΔG = ΔH – ΔTS  (8)

where G is the Gibbs free energy and H is enthalpy ≈ internal or mental energy, the function of intellect, creativity, and mental flexibility.

In exothermic changes ΔH < 0 or ΔTS > 0. However, endothermic process (ΔH > 0) can occur in supportive environments.                                                                    

The free energy principle and active inference can provide a mathematical argument for spontaneous processes, including emotional ones. The time-evolution of a system moves to thermodynamically more stable energy state (closer to thermodynamic equilibrium). The organism releases free energy in an exothermic process, which increases the entropy of the environment (e.g., criticism, anger, and brutality). Inversely, an endothermic process absorbs free energy from the environment. Therefore, emotions are part of a thermodynamic regulation of cognition, a fundamental part of the evolution of intellect.

Commonly, spontaneous behavior is called “free will” since it feels natural and easy. The exothermic cycle leads to irrational, damaging behavior, which, in extreme cases can lead to depression, disease, or the individual's demise.

One may conceive that hemispheric asymmetry could result in uneven thermodynamics between the brain hemispheres [132] but the structural analysis of intra- and inter-lobe connectivity is beyond the present work.

References from response to #2:

  1. Deli, J. Peters and A. Tozzi, "Relationships between short and fast brain timescales," Cognitive Neurodynamics, vol. 11, no. 539, 2017.

Deli, E., Peters, J., and Tozzi, A. (2018) The Thermodynamic Analysis of Neural Computation. J Neurosci Clin Res. 3:1.

  1. Deli, J. Peters and Z. Kisvarday, "The thermodynamics of cognition: A Mathematical Treatment," Computational and Structural Biotechnology Journal, no. https://doi.org/10.1016/j.csbj.2021.01.008, 2021.
  2. Déli and Z. Kisvárday, "The thermodynamic brain and the evolution of intellect: the role of mental energy," Cognitive neurodynamics, vol. 14, no. 6, pp. 743-756, 2020.

3) The proposed approaches are difficult to evaluate due to the lack of a reliable quantitative assessment.

We provided four arguments on page 6 (Section 3) supporting the significance of the thermodynamic cycles in the activation of the CNS 221-226:

  1. A stimulus triggers activation, which represents momentum and direction.

  1. Entropy-maximizing influences continuously adjust the large-scale spatial synchronization of oscillatory activity.
  2. The relaxation that recovers the resting state prepares the system for reactivation.
  3. Every cycle changes the brain's synaptic balance and organization.

To tackle the quantitative aspects in page 9 (Section 3.2), we produced a mathematical argument by utilizing the Kullback-Leibler divergence 326-372:

What amount of learning is required from the current knowledge p to reach knowledge q? When the distribution does not change with the change in the model parameters, the cost function is the Kullback-Leibler divergence ( ), a type of statistical distance or surprise [165] between the current output and the expected output [12, 13]. 

In the endothermic brain, the discrete is defined as follows:

X = discrete random variable in brain signal space X.

p(x) ≥ 0,q(x) > 0 = probability distribution of x, where

p(x) = probability distribution of observed brain signal x,

q(x) = probability distribution of estimated brain signal x.

 ≥ 0  (1)

where is the novelty of the information. Learning depends on ≥ 0 and when q = no learning occurs.

Instead of asking the amount of learning taking place between p and q, we can look for the cognitive updating caused by a stimulus, F (the free energy). S is the entropy, and T is the social temperature as defined by frequencies, dS = , therefore

F = <F> - TS  (2)

where <F> is the expected meaning (intellect). The system's free energy is proportional to the sensory information's surprise value relative to expectation. As the system moves toward equilibrium, the free energy is minimized. In the endothermic case, the resting entropy increases during stimulus processing, absorbing free energy.

The Boltzmann distribution expresses the probability that a system will be in a state   as a function of the intellect (energy) and the social temperature, showing an inverse relationship between aggravation (T) and intelligent behavior as follows.

  (4)

where k is the Boltzmann constant and the system evolves through a Markov process.

F(q) – F(p) = (5)

where F(q) is the free energy for q and F(p) is the free energy in p

From Eq. (2), we can express the energy requirement of learning

  (6)

  (7)

where is the average "relevance" of an incoming stimulus, is how much information is left to learn by going from p to q.

A similar argument was added in section 3.3 on P10. 400-424:

The free energy changes sign in Eq 1 and 2 because the low entropy state cannot absorb the energy of the stimulus. Although the exothermic state readily increases social temperature, aggravation disperses energy. In addition, anxiety fuels fear conditioning, a stronger, less flexible connectivity. Extensive psychological studies on stress and negative emotions support the above conclusion. Thus, a calm mind is most capable of learning and mental advancement.

Next, we use the Kullback-Leibler divergence for an exothermic situation.

Calculating  from Eq 1 in the exothermic case:

       ε = 0.01;

a = 3 / 5 – ε / 3;

b = 1 / 5 – ε / 3;

aa = 3 / 8 – ε / 3;

aa = 5 / 8 – ε / 3;

tl = a ln (a./ aa ) = 0.2828;

t2 = b ln(b./ bb) = -0.2251;

 = tl + t2 = 0.0577

Finally, we have addressed the minor comments below:

1) 144 link - the names of the authors are not indicated

Unfortunately, we were unable to find the requested link.

2) Links are not in the order of mention. For example: lines 209, 215, 224, 287, 404

P3-10, we made sure that the mentions in the text were in ascending order; 8 corrections were made.

3) Figure 1 is of poor quality. There is no link to figure 1 in the text of the article

P5. We removed Figure 1 from the manuscript.

4) It is not clear why exactly such geometric figures are chosen in Figure 3.

We inserted a more extensive explanation for Figure 3 (now Figure 2 on P5). P6. Top 203-208:

Figure 2. The analysis of neural activation as a harmonic motion The system returns to its starting position without creating memory (top). In a symbolic representation of neural activation, the stimulus pushes the neural system out of resting equilibrium (Bottom) Red color indicates the activation path between the brain’s modular structures forming a vector path (A-B) determined by the system's memory and the stimulus's energy.

We are thankful to the reviewer for his/her time and consideration of our work. We appreciate the helpful feedback, which we followed closely in our response.

Reviewer 2 Report (New Reviewer)

This is an interesting theoretical paper that attempts to explain emotions using a thermodynamic concept, the Carnot cycle. This paper is purely based on deductions from literature studies. It does not contain any results of simulations or experiments.

The problem of current form is that the novelty and evolution of this study by comparing recent literature is not clear. Thermodynamic concepts and their mathematics have already been applied in recent theoretical/computational neuroscience studies. For example, the free energy principle and active inference proposed by Friston et al. have provided conprehensive explanations of several cognitions including emotions. What is new compared to such literatures? How can proposed theory relate mathematically to such existing theories? At least, following literatures proposing emotion valence and arousal based on thermodynamic concepts should be cited and discussed for the mathematical/physical relations.

For valence:

Joffily, M., & Coricelli, G. (2013). Emotional valence and the free-energy principle. PLoS Computational Biology, 9(6), e1003094.

Hesp, C., Smith, R., Parr, T., Allen, M., Friston, K. J., & Ramstead, M. J. D. (2021). Deeply Felt Affect: The Emergence of Valence in Deep Active Inference. Neural Computation, 33(2), 398–446.

For arousal:

Yanagisawa, H. (2021). Free-Energy Model of Emotion Potential: Modeling Arousal Potential as Information Content Induced by Complexity and Novelty. Frontiers in Computational Neuroscience, 15, 107.

Technical comments:

Table 1 should be important. However, the text does not provide sufficient explanation.

Line 322: KL-divergence is not a distance by definition. The implications of KL divergence as novelty and surprise have been discussed. Related literatures should be cited and discussed. For example:

Itti, L., & Baldi, P. (2009). Bayesian surprise attracts human attention. Vision Research, 49(10), 1295–1306.

Yanagisawa, H., Kawamata, O., & Ueda, K. (2019). Modeling Emotions Associated With Novelty at Variable Uncertainty Levels: A Bayesian Approach. Frontiers in Computational Neuroscience13(2). 

Line 325-327: the almost same formula repeats. All formula should be numbered.

For 3.3, a mathematical analysis like in 3.2 is desired to clearly understand the relationship with emotions.

Author Response

We want to thank the reviewer for the input on the manuscript: "HOW THE BRAIN BECOMES THE MIND: CAN THE CARNOT CYCLE EXPLAIN THE EMERGENCE AND NATURE OF EMOTIONS?" please find our point by point response to the feedback in red and the revised text in italics below.

This is an interesting theoretical paper that attempts to explain emotions using a thermodynamic concept, the Carnot cycle. This paper is purely based on deductions from literature studies. It does not contain any results of simulations or experiments.

The problem of current form is that the novelty and evolution of this study by comparing recent literature is not clear. Thermodynamic concepts and their mathematics have already been applied in recent theoretical/computational neuroscience studies. For example, the free energy principle and active inference proposed by Friston et al. have provided conprehensive explanations of several cognitions including emotions. What is new compared to such literatures? How can proposed theory relate mathematically to such existing theories? At least, following literatures proposing emotion valence and arousal based on thermodynamic concepts should be cited and discussed for the mathematical/physical relations.

1, The problem of current form is that the novelty and evolution of this study by comparing recent literature is not clear.

The novelty of the present work is demonstrating that resting entropy is a measure of thermodynamic free energy, which regulates stimulus-response and spontaneous behavior. Indeed, high resting entropy permits synaptic flexibility and complex behavior, but low entropy expels heat via criticism and aggression, engendering insecurity and regret. Moreover, passive information accumulation may compromise hormonal regulation.

In the Intrioduction P1. 35-37:

Although the scientific understanding of emotions has been challenging, their nature is of utmost importance in psychology, education and AI research.  

In Conclusions P12. 2nd paragraph 501-503:

We used the free energy principle to analyze emotions and spontaneous behavior. We show that resting entropy is a measure of the thermodynamic free energy, regulating stimulus response and spontaneous behavior.

In Conclusions, P12. 4th paragraph 513-515:

The thermodynamic analysis of the brain's evoked cycle shows that emotions have action-producing power.

2) Thermodynamic concepts and their mathematics have already been applied in recent theoretical/computational neuroscience studies. For example, the free energy principle and active inference proposed by Friston et al. have provided conprehensive explanations of several cognitions including emotions. What is new compared to such literatures? How can proposed theory relate mathematically to such existing theories?

The Free Energy Principle (FEP) is currently one of the most promising frameworks which offers a unified explanation to a wide range of systems and life-related phenomena. However, despite of its wide applicability, it lacks falsifiable predictions.

In Abstract, on P1. 21-22:

Our work provides an analytical perspective of positive and negative emotions and spontaneous behavior starting from the free energy principle.

Our conclusions offer falsifiable predictions as summarized in Table II.  on P12. 491-492:

Analysis of spontaneous behavior

Exothermic reaction (mental energy loss)

Endothermic reaction (mental energy gain)

High entropy environment (stress)

Spontaneous behavior

Spontaneous on low social temperature, which permits overcoming the negativity

Low entropy, supportive environment

Spontaneous on high social temperature because the support cannot calm the aggravation

Spontaneous behavior

Table II. Analysis of behavior as a function of environmental and personal factors

Also on P10-11, starting on P10 bottom 428-442.

How can information accumulation lead to adverse psychological symptoms?

  1. Like all exothermic processes [54, 55, 56], the low entropy brain loses energy (Table I). For example, it dissipates energy to the environment via criticism, destructive behavior, or violence [7, 8].
  2. However, in contrast to physical processes, the reversible brain activations stabilize low entropy, creating long-term adverse emotional and psychological outcomes.
  3. Long-term potentiation reduces the degrees of freedom due to the loss of synaptic complexity, producing repetitious, monotone thinking [118]. The negative thought pattern becomes more powerful and pessimistic through a Bayesian process, affecting behavior. For example, the severity of cognitive impairment in depression correlates with brain entropy reduction [118].
  4. Attention to the past degrades optimism for the future [119, 120]. In addition, the energy imbalance may misalign hormone regulation with adverse mental and immune consequences, anxiety, and depression [75, 117].

3) At least, following literatures proposing emotion valence and arousal based on thermodynamic concepts should be cited and discussed for the mathematical/physical relations.

In Introduction, on P2-3, starting on P2 bottom.95-97:

In our discussion of emotions and their thermodynamic consequences, we follow earlier insightful work on the free energy principle and the thermodynamic analysis of cognition [22, 23, 24, 25, 26].

On P10, Section 3.3, 1st paragraph 400-405.

Energy generation by exothermic processes is the basis of the second law of thermodynamics. The exothermic cycle in sensory processing might have emerged to correct mistakes; the feeling that “things did not go well” inspires changes during action preparation and execution. Nevertheless, the exothermic cycle changes cortical representations, which can lead to anxiety [116], apathy, and depression [75, 117], via a Bayesian process (Figure 4).

In addition, we added a section (Section 4) on spontaneous behavior, providing elaboration via the free energy principle (p11-12) 463-490.:

Spontaneous processes

Most cognitive and behavioral actions represent a response to a stimulus, but spontaneous actions, although often precipitated by a related stimulus, show no relationship to its nature or magnitude. For example, a gunman’s shooting spree can be triggered by a job loss, but it results from long-term insecurity and hopelessness.

In thermodynamics, a spontaneous process transpires without external input to the system (Table II) which can be assessed by the Gibbs free energy equation. 

ΔG = ΔH – ΔTS  (8)

where G is the Gibbs free energy and H is enthalpy ≈ internal or mental energy, the function of intellect, creativity, and mental flexibility.

In exothermic changes ΔH < 0 or ΔTS > 0. However, endothermic process (ΔH > 0) can occur in supportive environments.                                                                    

The free energy principle and active inference can provide a mathematical argument for spontaneous processes, including emotional ones. The time-evolution of a system moves to thermodynamically more stable energy state (closer to thermodynamic equilibrium). The organism releases free energy in an exothermic process, which increases the entropy of the environment (e.g., criticism, anger, and brutality). Inversely, an endothermic process absorbs free energy from the environment. Therefore, emotions are part of a thermodynamic regulation of cognition, a fundamental part of the evolution of intellect.

Commonly, spontaneous behavior is called “free will” since it feels natural and easy. The exothermic cycle leads to irrational, damaging behavior, which, in extreme cases can lead to depression, disease, or the individual's demise.

One may conceive that hemispheric asymmetry could result in uneven thermodynamics between the brain hemispheres [132] but the structural analysis of intra- and inter-lobe connectivity is beyond the present work.

We discuss the mathematical/physical relations of the endothermic cycle, Section 3.2, on P9. 326-372:

What amount of learning is required from the current knowledge p to reach knowledge q? When the distribution does not change with the change in the model parameters, the cost function is the Kullback-Leibler divergence ( ), a type of statistical distance or surprise [165] between the current output and the expected output [12, 13]. 

In the endothermic brain, the discrete  is defined as follows:

X = discrete random variable in brain signal space X.

p(x) ≥ 0,q(x) > 0 = probability distribution of x, where

p(x) = probability distribution of observed brain signal x,

q(x) = probability distribution of estimated brain signal x.

 ≥ 0  (1)

where is the novelty of the information. Learning depends on ≥ 0 and when q = no learning occurs.

Instead of asking the amount of learning taking place between p and q, we can look for the cognitive updating caused by a stimulus, F (the free energy). S is the entropy, and T is the social temperature as defined by frequencies, dS = , therefore

F = <F> - TS  (2)

where <F> is the expected meaning (intellect). The system's free energy is proportional to the sensory information's surprise value relative to expectation. As the system moves toward equilibrium, the free energy is minimized. In the endothermic case, the resting entropy increases during stimulus processing, absorbing free energy.

The Boltzmann distribution expresses the probability that a system will be in a state   as a function of the intellect (energy) and the social temperature, showing an inverse relationship between aggravation (T) and intelligent behavior as follows.

  (4)

where k is the Boltzmann constant and the system evolves through a Markov process.

F(q) – F(p) = (5)

where F(q) is the free energy for q and F(p) is the free energy in p

From Eq. (2), we can express the energy requirement of learning

  (6)

  (7)

where is the average "relevance" of an incoming stimulus, is how much information is left to learn by going from p to q.

Next, we use the Kullback-Leibler divergence for an exothermic situation, Senction 3.3, P10. 400-424:

Calculating     from Eq 1 in the exothermic case:

       ε = 0.01;

a = 3 / 5 – ε / 3;

b = 1 / 5 – ε / 3;

aa = 3 / 8 – ε / 3;

aa = 5 / 8 – ε / 3;

tl = a ln (a./ aa ) = 0.2828;

t2 = b ln(b./ bb) = -0.2251;

 = tl + t2 = 0.0577

We made the following corrections in the references:

For valence:

3a) P7. We inserted the following two references.[89,90] 264:

Joffily, M., & Coricelli, G. (2013). Emotional valence and the free-energy principle. PLoS Computational Biology9(6), e1003094.

Hesp, C., Smith, R., Parr, T., Allen, M., Friston, K. J., & Ramstead, M. J. D. (2021). Deeply Felt Affect: The Emergence of Valence in Deep Active Inference. Neural Computation33(2), 398–446.

For arousal:

P11. We inserted the following reference [131] 461.

Yanagisawa, H. (2021). Free-Energy Model of Emotion Potential: Modeling Arousal Potential as Information Content Induced by Complexity and Novelty. Frontiers in Computational Neuroscience15, 107.

3b) Technical comments:

Table 1 should be important. However, the text does not provide sufficient explanation.

P 4, we made the following correction to the table I.155-159:

Physical systems

Brain activations

Microstates orientation

Oriented in space

Information entropy oriented in time

System evolution

Brownian motion

Wave-like activations founded on memories

Entropic force

Irreversible macroscopic behavior

Irreversible activations

The consequences of irreversibility

The arrow of time

Future orientation, novelty, curiosity, and creativity

High entropy state

Equilibrium

Equilibrium

−−−−− Consequences of high entropy

Loss of work potential

Intellect, confidence, and a can-do attitude

−−−−− Energy input lowers the entropy

The system moves away from equilibrium, but irreversibility remains!

Reversible and repetitive activations

−−−−− Consequences of low entropy

Increasing work potential

Uncertainty, lack of control, and psychological problems

Table I. A systematic comparison of the thermodynamics of physical and neural systems The top three rows report on the orientation, evolution, and directionality of the system’s microstates. The lowest four rows the consequences of the above differences in the microstates. 

3c) Line 322: KL-divergence is not a distance by definition. The implications of KL divergence as novelty and surprise have been discussed. Related literatures should be cited and discussed. For example:

We corrected our argument indicating that KL divergence is a statistical distance between the current output of the algorithm and the expected output. We also inserted the two suggested references p2,9. [12,13]

In Section 3.2, on P9, top. 327-329:

When the distribution does not change with the change in the model parameters, the cost function is the Kullback-Leibler divergence ( ), a statistical distance or surprise [165] between the current output and the expected output [12, 13]. 

Itti, L., & Baldi, P. (2009). Bayesian surprise attracts human attention. Vision Research49(10), 1295–1306.

Yanagisawa, H., Kawamata, O., & Ueda, K. (2019). Modeling Emotions Associated With Novelty at Variable Uncertainty Levels: A Bayesian Approach. Frontiers in Computational Neuroscience13(2). 

Line 325-327: the almost same formula repeats. All formula should be numbered.

P 9. We numbered all the formulas through Line 336, 346, 356, 360, 367, 368, 471.

3d) For 3.3, a mathematical analysis like in 3.2 is desired to clearly understand the relationship with emotions.

P 10. We inserted a mathematical analysis to Section 3.3. 406-424.

The free energy changes sign in Eq 1 and 2.

From -F  (1)

therefore,                                    

The free energy changes sign in Eq 1 and 2, because the low entropy state cannot absorb the energy of the stimulus. Although the exothermic state readily increases social temperature, aggravation disperses energy. In addition, anxiety fuels fear conditioning, a stronger, less flexible connectivity. Extensive psychological studies on stress and negative emotions support the above conclusion. Thus, a calm mind is most capable of learning and mental advancement.

Next, we use the Kullback-Leibler divergence for an exothermic situation.

Calculating  from Eq 1 in the exothermic case:

       ε = 0.01;

a = 3 / 5 – ε / 3;

b = 1 / 5 – ε / 3;

aa = 3 / 8 – ε / 3;

aa = 5 / 8 – ε / 3;

tl = a ln (a./ aa ) = 0.2828;

t2 = b ln(b./ bb) = -0.2251;

 = tl + t2 = 0.0577

We are thankful to the referee for the fair but thorough comments, which resulted in a better-focused and more relevant manuscript.

Reviewer 3 Report (New Reviewer)

Indeed “The neural systems' electric activities are fundamental for the phenomenology of consciousness” In the bilateral human brain all electrical, and biochemical activities related to the sensory perception exhibit left-right- hemispheric asymmetry (see for example [Jao et al. 2020]).  Correspondently all thermodynamic parameters, including entropy, should be considered in the view of brain hemispheric asymmetry. All paper ignoring this request must be rejected.

[Jao et al. 2020] Chi-Wen Jao, Jiann-Horng Yeh, Yu-Te Wu, et al. Alteration of the Intra- and Inter-Lobe Connectivity of the Brain Structural Network in Normal Aging. Entropy 2020, 22(8), 826, doi.org/10.3390/e22080826

Author Response

We want to thank the reviewer for the input on the manuscript: "HOW THE BRAIN BECOMES THE MIND: CAN THE CARNOT CYCLE EXPLAIN THE EMERGENCE AND NATURE OF EMOTIONS?" Please find our point by point response to the feedback in red and the revised text in italics below.

Indeed “The neural systems' electric activities are fundamental for the phenomenology of consciousness” In the bilateral human brain all electrical, and biochemical activities related to the sensory perception exhibit left-right- hemispheric asymmetry (see for example [Jao et al. 2020]).  Correspondently all thermodynamic parameters, including entropy, should be considered in the view of brain hemispheric asymmetry. All paper ignoring this request must be rejected.

Thank you for suggesting the consideration of the brain’s hemispheric asymmetry of entropy. However, our work focuses on ideal cognitive performance. In the literature, the brain’s entropic hemispheric asymmetry is a controversial topic. Some authors observed significant hemispheric entropy differences in normal brain function (Alù et al., 2020), while others confirmed a symmetry (Li and Zhang, 2022; Vecchio et al., 2021). Yet others associates hemispheric asymmetries to particular task (Zhang et al., 2005) or various brain diseases, such as Parkinson’s disease (Li et al., 2020), bipolar disorder (Wang et al., 2018), Alzheimer's disease, finding pronounced heritability (Lubben et al., 2021). Based on the leftward dominance for speech, some scientists point to the dominance of the right hemisphere in overall functioning (Bisiacchi and Cainelli, 2022), gradually diminishing during the aging process (Jao et al., 2021; Knights et al., 2021). Nevertheless, most authors study brain entropy, independent of hemispheric asymmetry considerations, support our conclusions (Keshmiri, 2020).

With the above considerations, we did not feel that the brain’s hemispheric asymmetry, albeit an area of investigation with great significance, is pertinent to our discussion.

References from the above feedback:

Alù F, Miraglia F, Orticoni A, Judica E, Cotelli M, Rossini PM, Vecchio F. Approximate Entropy of Brain Network in the Study of Hemispheric Differences. Entropy (Basel). 2020 Oct 27;22(11):1220.

Bisiacchi, P., Cainelli, E. Structural and functional brain asymmetries in the early phases of life: a scoping review. Brain Struct Funct 227, 479–496 (2022).

Chi-Wen Jao, Jiann-Horng Yeh, Yu-Te Wu, et al. Alteration of the Intra- and Inter-Lobe Connectivity of the Brain Structural Network in Normal Aging. Entropy 2020, 22(8),

Keshmiri S. Entropy and the Brain: An Overview. Entropy (Basel). 2020 Aug 21;22(9):917. 

Knights, E., Morcom, A.M., & Henson, R.N. (2021). Does Hemispheric Asymmetry Reduction in Older Adults in Motor Cortex Reflect Compensation? The Journal of Neuroscience, 41, 9361 - 9373.

Li, P., Ensink, E., Lang, S. et al. Hemispheric asymmetry in the human brain and in Parkinson’s disease is linked to divergent epigenetic patterns in neurons. Genome Biol 21, 61 (2020).

Li, Y., & Zhang, Z. (2022). Enhanced Laterality Index: A Novel Measure for Hemispheric Asymmetry. Journal of Healthcare Engineering, 2022.

Lubben, N., Ensink, E., Coetzee, G. A., Labrie, V. The enigma and implications of brain hemispheric asymmetry in neurodegenerative diseases, Brain Communications, Volume 3, Issue 3, 2021.

Vecchio, F.; Miraglia, F.; Pappalettera, C.; Orticoni, A.; Alù, F.; Judica, E.; Cotelli, M.; Rossini, P.M. Entropy as Measure of Brain Networks’ Complexity in Eyes Open and Closed Conditions. Symmetry 2021, 13, 2178.

Wang, B., Li, T., Zhou, M., Zhao, S., Niu, Y., Wang, X., Yan, T., Cao, R., Xiang, J., & Li, D. (2018). The Abnormality of Topological Asymmetry in Hemispheric Brain Anatomical Networks in Bipolar Disorder. Frontiers in Neuroscience, 12.

Zhang, L., & Zheng, C. (2005). Cortical Lateralization Analysis by Kolmogorov Entropy of EEG. Panhellenic Conference on Informatics.

P 12. We inserted the reference suggested by the reviewer in the manuscript, line 489:

One may conceive that hemispheric asymmetry could result in uneven thermodynamics between the brain hemispheres [132] but the structural analysis of intra- and inter-lobe connectivity is beyond the present work.

  1. Jao et al. 2020]Chi-Wen Jao, Jiann-Horng Yeh, Yu-Te Wu, et al. Alteration of the Intra- and Inter-Lobe Connectivity of the Brain Structural Network in Normal Aging. Entropy 2020, 22(8), 826, doi.org/10.3390/e22080826

We are thankful to the reviewer for the consideration of our work. We appreciate for bringing to our attention the importance of the brain’s hemispheric asymmetry. We hope to address those concerns in future work. We hope that our argument is convincing and can facilitate the publication of this important and relevant manuscript.

Round 2

Reviewer 1 Report (New Reviewer)

I thank the respected editors and authors for the work done. I have several questions about this article.

1) The article contains a large number of logical arguments. The results of practical measurements are not given in this article. The expression “ the thermodynamic analysis may ascertain the brain's computational power ” (line 517) cannot be applied in practice without experimental data.

2) Line 329 is a reference to a literary source [165], and there are 132 items in the list of references.

3) It is not clear for what purpose the calculation in lines 417-424 is given.

Unfortunately, I think that this manuscript should not be published in a journal “Entropy”. This article is probably more suitable for specialized medical and social life journals.

Author Response

We want to thank the reviewer for the input on the manuscript: "HOW THE BRAIN BECOMES THE MIND: CAN THE CARNOT CYCLE EXPLAIN THE EMERGENCE AND NATURE OF EMOTIONS?" Please find our point-by-point response in red and the text to be inserted in the paper in red italics below.

1) The article contains a large number of logical arguments. The results of practical measurements are not given in this article. The expression “ the thermodynamic analysis may ascertain the brain's computational power ” (line 517) cannot be applied in practice without experimental data.

We agree with the Reviewer and removed the statement in Conclusions:

P13, Conclusions: “the thermodynamic analysis may ascertain the brain's computational power ” (line 517)”

Regarding the remark "The results of practical measurements are not given in this article" our motivation is guided by the expectation that the overall energy availability in the brain is thought to be constant but can display substantial local differences. Although imaging technologies such as EEG, CT scan, PET, MRI, or near-infrared spectroscopy (NIRS) can be used to measure the brain energy metabolism, they do not provide sufficient temporal or spatial resolution. Therefore, we added the following text regarding possible studies to measure the brain’s energy balance on page 6-7 and 12:

P 6-7: The overall energy availability in the brain is thought to be constant but can display substantial local differences, which can be used to measuring energy utilization during stressful and positive conditions by simultaneous EEG-fMRI analysis. Other studies might include broadband near-infrared spectroscopy, which can measure energy metabolism in brain cells’ mitochondria [81].

P12: Nevertheless, numerous publications support the contrasting metabolisms positive and negative emotions [81, 89, 90, 91].

Added in the reference list:

  1. Bale. G, Mitre. S, Meek. J and et al., "A new broadband near-infrared spectroscopy system for in-vivo measurements of cerebral cytochrome-c-oxidase changes in neonatal brain injury," Biomed Opt Express, vol. 5, no. 10, pp. 3450-66, 2014.
  2. Z. Gao, X. Cui, W. Wan, W. Zheng and Z. Gu, "Long-range correlation analysis of high frequency prefrontal electroencephalogram oscillations for dynamic emotion recognition," Biomedical Signal Processing and Control, 2022.
  3. C. Hesp, R. Smith, T. Parr, M. Allen, K. Friston and M. Ramstead, "Deeply Felt Affect: The Emergence of Valence in Deep Active Inference," Neural Computation, vol. 33, pp. 1-49, 2021.
  4. M. Joffily and G. Coricelli, "Emotional valence and the free-energy principle," PLoS Computational Biology, vol. 9, no. 6 , p. e1003094., 2013.

2) Line 329 is a reference to a literary source [165], and there are 132 items in the list of references.

The earlier version of the manuscript contained 165 reference items. During revisions, that number shrunk to132 but accidentally reference # 165 remained in the final version. We are thankful for the Reviewer for noticing our mistake, which was corrected.

3) It is not clear for what purpose the calculation in lines 417-424 is given.

We appreciate the reviewer’s note and left out the calculation. For clarity, we decided to follow the argument introduced for the endothermic brain and modified the text on page 10:

The energy state and the entropy of the exothermic brain are smaller than the endothermic condition. Equation (2) shows that high social temperature can turn the free energy negative. Therefore, the free energy changes sign in Eq 1 and 2, because the low entropy state cannot absorb the energy of the stimulus. Although the exothermic state readily increases social temperature, aggravation disperses energy. In addition, anxiety fuels fear conditioning, a stronger, less flexible connectivity. Extensive psychological studies on stress and negative emotions support the above conclusion. Even though the high mental energy brain tends to form an endothermic cycle, high social temperature can produce an exothermic cycle. Thus, a calm mind is most capable of learning and mental advancement.

The exothermic equilibrium is not a high entropy state; the brain responds to even minor difficulty by increasing social temperature (excitement or aggravation): TS  >>  <F>, which dramatically changes the cycle’s dynamics. Because the high entropy state occurs early in the cycle, without time for contemplation, the high social temperature fuels energy loss by urgent, thoughtless actions. According to the thermodynamic principles, Efficiency  =  =  where is the social temperature in the resting state,  is the social temperature during the first phase of the cycle.

Therefore, the exothermic brain converts heat (QH) to work (W): W =  QH.

4) Unfortunately, I think that this manuscript should not be published in a journal “Entropy”. This article is probably more suitable for specialized medical and social life journals.

Entropy, a fundamental concept in thermodynamics, measures the energy unavailable to do physical work in a physical system. The concept was adopted in information theory, sociology, and economy. The journal Entropy even has a special section dedicated to “Entropy and Biology.” Recently, brain entropy has become an important indicator of cognitive ability, intellectual prowess, and certain brain diseases. The importance and respect of this field is indicated by Entropy’s recent Special Issues: Brain Connectivity Complex Systems (open); Entropy on Biosignals and Intelligent Systems(closed 30 September 2022), Entropy Applications in Electroencephalography (open), Information Theory in Neuroscience (closed 30 April 2018), and Information Theory in Neural Coding and Decoding (closed 30 September 2022).

In 2017, the seminal work of Robert Fry’s “Physical Intelligence and Thermodynamic Computing,” published in the journal Entropy, started the thermodynamic investigation of the neural system. We feel obliged to continue the topic in the same journal with expert review and readership.

We are thankful to the reviewer for his/her time and consideration of our work. We appreciate the helpful feedback, which we followed closely in our response.

Reviewer 2 Report (New Reviewer)

Page 10: Please explicitly explain what ε, a, b, aa, t1, and t2 are and how these parameters are obtained from Eq. 1 which does not include there parameters. Also, aa is defined twice. I don't understand why you calcurated the DKL with numbers since there are no discussions about the calculation results. Please contextualize the DKL calculations.

Author Response

We want to thank the reviewer for the input on the manuscript: "HOW THE BRAIN BECOMES THE MIND: CAN THE CARNOT CYCLE EXPLAIN THE EMERGENCE AND NATURE OF EMOTIONS?" Please find our response in red below and the text to be inserted in the paper in red italics.

Page 10: Please explicitly explain what ε, a, b, aa, t1, and t2 are and how these parameters are obtained from Eq. 1 which does not include there parameters. Also, aa is defined twice. I don't understand why you calcurated the DKL with numbers since there are no discussions about the calculation results. Please contextualize the DKL calculations.

We acknowledge that our argument was confusing and not well thought out. Therefore, we decided to follow the thread introduced in the endothermic perception and modified the following text.

P11. The energy state and the entropy of the exothermic brain are smaller than the endothermic condition. Equation (2) shows that high social temperature can turn the free energy negative. Therefore, the free energy changes sign in Eq 1 and 2, because the low entropy state cannot absorb the energy of the stimulus. Although the exothermic state readily increases social temperature, aggravation disperses energy. In addition, anxiety fuels fear conditioning, a stronger, less flexible connectivity. Extensive psychological studies on stress and negative emotions support the above conclusion. Even though the high mental energy brain tends to form an endothermic cycle, high social temperature can produce an exothermic cycle. Thus, a calm mind is most capable of learning and mental advancement.

The exothermic equilibrium is not a high entropy state; the brain responds to even minor difficulty by increasing social temperature (excitement or aggravation): TS  >>  <F>, which dramatically changes the cycle’s dynamics. Because the high entropy state occurs early in the cycle, without time for contemplation, the high social temperature fuels energy loss by urgent, thoughtless actions. According to the thermodynamic principles, Efficiency  =  =   where is the social temperature in the resting state,  is the social temperature during the first phase of the cycle.

Therefore, the exothermic brain converts heat (QH) to work (W):W =  QH.

We are thankful to the reviewer for his/her time and consideration of our work. We appreciate the helpful feedback, which we followed closely in our response.

Reviewer 3 Report (New Reviewer)

The structural and functional analysis of intra- and inter-lobe connectivity and corresponding handedness are beyond the present work. This fact significantly narrow the predictive power of conclusions.

Author Response

We want to thank the reviewer for the input on the manuscript: "HOW THE BRAIN BECOMES THE MIND: CAN THE CARNOT CYCLE EXPLAIN THE EMERGENCE AND NATURE OF EMOTIONS?" Please find our response in red below.

The structural and functional analysis of intra- and inter-lobe connectivity and corresponding handedness are beyond the present work. This fact significantly narrow the predictive power of conclusions.

Conscious science is an emerging science with uncertainties surrounding even the most basic questions, such as duality or the hard problem of consciousness. A recent google search found only about 0.5 % cognitive science manuscripts focusing on brain hemispheric asymmetry. Therefore, at this early stage of research, a discussion on hemispheric asymmetry should not be a publication requirement. We hope that our response reflects our respect of the Reviewer’s feedback.

We are thankful again to the Reviewer for his/her time and consideration of our work without questioning the importance of hemispheric asymmetry studies in cognitive and brain science.

Round 3

Reviewer 1 Report (New Reviewer)

I thank the editor and authors for their work. I think that the article has become much better than it was at the beginning. If the authors improve the quality of the figures in the article, then the article can be accepted for publication.

This manuscript is a resubmission of an earlier submission. The following is a list of the peer review reports and author responses from that submission.

Round 1

Reviewer 1 Report

This paper presents a model in which emotions arise from the thermodynamic organization of the brain. The theoretical development comprises multiple parts. In the first part (“Mental unity”), makes some preliminary remarks about consciousness, cortico-thalamic interactions, synaptic maps, electrical activity of the brain and other topics. The second part (“The resting state…”), argues that the resting-state activity of the brain underlies the conscious sense of self that people (and maybe other animals) have. The main idea of the paper is that each incoming stimulus perturbs the brain from its resting state and that evokes a thermodynamic cycle (a Carnot Cycle) that restores it to its resting state. The cycle can be endothermic or exothermic and can produce or consume entropy. Positive and negative emotions are related to the actions of this evoked cycle.

This paper contributes worthwhile original ideas for relating mental functions to the physical (in particular thermodynamic) actions of the brain. It requires, however, substantial revision. The paper should be rewritten with better organization so that the ideas flow step-by-step. More concrete explanations are needed. The authors go off on numerous tangents that are not clearly related to the main themes. This should be cut way back. The authors should avoid excursions and stay focused on developing the major topics.

Some more specific critiques follow.

  1. The authors refer to “consciousness” and then shortly thereafter to “self-consciousness”. That might confuse some readers. Instead of the latter, perhaps they should say “sense of self”.

  1. The statement, “The sensory system projects the physical environment into the brain via a passive process” is not correct. Even in the strict thermodynamic sense, perception is an active process, i.e., it consumes energy. In the retina, for example, even in complete darkness, there runs a so-called dark current, which consumes energy. All the more when the retina is bombarded by photons. In the more general behavioral sense, perception is also an active process. We are constantly making active decisions about which way to avert the eyes, which way to turn the head for looking and/or listening, which surfaces to palpate with the fingertips, etc., etc. The authors should revise their model to accommodate energetic needs of perceptual processes by

  1. The statement, “Furthermore, the inability to consciously guide behavior leads to remorse, shame, and regret,” is also off the mark. Remorse, or guilt, happens not when one is unable to guide behavior due to emotions, but when one-- tempted by emotions-- consciously does something that turns out bad. E.g., “I shoplifted and I got caught. Now I have a criminal record.” Regret happens when one consciously does not do something that one could have and should have done. E.g., “I should have thrown the life preserver to the drowning man, but I didn’t. Now he’s dead and his family is blaming me.” Shame is not even on this spectrum. Shame is not about what one did, but about who one is. E.g., “The pauper was ashamed to show-up in rags at the wealthy couple’s wedding.”

  1. Related to 3., the paper largely takes the point of view that free will doesn’t exist because emotions irrepressibly drive behavior. In fact they say directly, “The impression of free will masks emotions' irresistible power over behavior.” It is acceptable to express this point of view, but it would be a good idea to mention some alternatives. One alternative is that volition really does exist and plays some role in our behavior. Another is that emotions do not drive behavior, bur are what occurs in the mind AFTER behavior has been decided.

  1. Some abbreviations are not defined, e.g., “DNN”.

  1. One aspect of the theory is that “Sensory reality is spatial”. Then the authors say that the spatial configuration of reality gets recoded by the brain into a temporal series of activity. But sensory reality is itself is temporal as well as spatial. The brain has systems explicitly for the perception of motion. This should be worked into the model.

  1. “The endothermic cycle is an energetic, high entropy state.” That sounds like loose terminology. A cycle is not a state. Systems have states, not cycles. A cycle moves a system between states.

  1. Is Eq. (3) “cosα2 + sinα2=1” in error? Should it be “cos2α + sin2α=1” (i.e., (cosα)2 + (sinα)2=1)? The former equation is in principle possible, but the latter would be much, much more common.

  1. Is Eq. (7) “?0=∮???=∮??? =dT− dS” in error? As written, it does not look dimensionally consistent. dT has units of K, dS has units of J/K and W0, ∮??? and ∮??? have units of J. So, unless it’s written in dimensionless variables there must be one or more errors.

  1. “Temperature is the source of heat and the manifestation of thermal energy [116], which flows from a higher to a lower level [117].” This is inaccurate and/or awkward. “Heat” and “thermal energy” are more or less synonyms. They both refer to energy that has not been given detailed accounting. E.g., to the translations, rotations, vibrations, etc. of molecules whose trajectories are not being observed or recorded directly, but rather are amalgamated non-specifically. “Temperature”, at least in common thermodynamic usage, is a measure of the distribution of this thermal energy amongst the molecules or other particles that constitute the system in question. It is proportional to the width of the distribution, something like a standard deviation in statistics. But temperature is not the source of heat. A heat source is either a system of any kind that already contains this thermal energy at a higher temperature than some other system with which it is in contact, or a system containing energy in some higher form (e.g., mechanical energy, electrical energy, electromagnetic radiation,…) that is explicitly accounted for but that is being degraded into thermal energy.

There are further issues in the paper similar to those described above that require attention.

Author Response

Thank you for giving us the opportunity to submit a revised draft of our manuscript titled "HOW THE BRAIN BECOMES THE MIND: CAN THE BORN RULE AND THE CARNOT CYCLE EXPLAIN THE COGNITIVE PROCESS?" to the journal Entropy. We are grateful to the reviewer for the insightful comments. We have been able to incorporate changes to reflect most of the suggestions provided by the reviewers. We have used blue highlight in our response below, but red within the manuscript to highlight the changes.

 1, "self-consciousness". might confuse some readers perhaps should say "sense of self".

We have changed self-consciousness into sense of self. Three changes were made in the document.

2, The sensory system projects the physical environment into the brain via a passive process" is not correct. perception is also an active process. 

We made the correction p2: "The sensory system projects the physical environment into the brain via an active, energy requiring process, often independent of conscious intention. For example, environmental noise, disturbing light or noxious fumes can intrude on current activity, even sleep."

3, "Furthermore, the inability to consciously guide behavior leads to remorse, shame, and regret," is also off the mark. Remorse, or guilt, happens not when one is unable to guide behavior due to emotions, but when one-- tempted by emotions-- consciously does something that turns out bad. E.g., "I shoplifted and I got caught. 

We followed the suggestions regarding the role of conscious guidance of behavior. P 7. It was impotent to make a distinction, because emotions can be controlled in normal daily life, but occasionally, they get out of control. For example, remorse, or guilt, happens not when one-- tempted by emotions-- consciously does something that turns out bad. E.g., "I shoplifted, and I got caught. In turn, remorse or guilt over mistakes leads to involuntary emotions, which require mental or physical effort to change [95, 96, 97].

P 2. "Even when mistakes, transgressions are occur due to conscious decisions, the resulting remorse, shame, and regret are beyond conscious control. In other words, remorse, shame, and regret takes time and struggle to settle." In extreme cases, such as PTSD, anxiety and depression, emotions have irresistible power over behavior.

4, The impression of free will masks emotions' irresistible power over behavior." It is acceptable to express this point of view, but it would be a good idea to mention some alternatives. One alternative is that volition really does exist and plays some role in our behavior. Another is that emotions do not drive behavior, bur are what occurs in the mind AFTER behavior has been decided.

We inserted correction into the text on p7.

For example, post-traumatic stress disorder (PSTD) can completely rule thinking. Furthermore, conscious mistakes and transgressions often lead to remorse, shame, and regret that are beyond conscious control. Like gravitational imbalance, threats to personal security or ego triggers balancing measures, which can dominate thinking. Therefore, free will is graded ability, corrupted by depression and anxiety.

We also inserted two new figures (Figure 3 and 4) to better illustrate emotions energy nature, based on new work by Gao et al. (2022).

5, Some abbreviations are not defined, e.g., "DNN".

We defined abbreviations used in the text. 

6, "Sensory reality is spatial". Then the authors say that the spatial configuration of reality gets recoded by the brain into a temporal series of activity. But sensory reality is itself is temporal as well as spatial. The brain has systems explicitly for the perception of motion. This should be worked into the model.

We interpreted and defined the relationship between sensory reality and representation in the brain. P 5.

For example, the description of an object requires a temporal order of speech, writing, poetry, or music and can recreate the original object as a mental reflection. Furthermore, Kant [58] has already pointed out that life's biological dependence on air, water, rest, and food dictates a temporal organization. Therefore, the idea of space-time resurfaces in the brain with a more significant temporal component. Thus, the temporal organization of neural processing supports neural changes during task-switching [63, 60], and the order associations in learning, speech, thinking, and muscle coordination are fundamental temporal in nature [64, 59].

7, "The endothermic cycle is an energetic, high entropy state." That sounds like loose terminology. A cycle is not a state. Systems have states, not cycles. A cycle moves a system between states.

We made the requested correction, for example, on p 8 - 10.

We applied the reversible Carnot cycle to characterize an ideal energy/information cycle between two cognitive states with vanishing net entropy production. The cycle's energy gain is conserved in the synaptic map, turning it into an "invisible" factor in cognition structural relationships. The synaptic connections form an "independent medium," allowing the cycle to repeat with the same state variables of entropy as a complexity-increasing harmonic motion.

8, Is Eq. (3) “cosα2 + sinα2=1” in error? Should it be “cos2α + sin2α=1” (i.e., (cosα)2 + (sinα)2=1)? The former equation is in principle possible, but the latter would be much, much more common

We appreciate the suggestion and corrected the error, p 8.

 (5) Some abbreviations are not defined, e.g., "DNN".

We defined abbreviations used in the text.  

9, Is Eq. (7) "?0=∮???=∮??? =dT− dS" in error? As written, it does not look dimensionally consistent. dT has units of K, dS has units of J/K and W0, ∮??? and ∮??? have units of J. So, unless it's written in dimensionless variables there must be one or more errors.

We appreciate the reviewer's attention to detail. We made the appropriate correction.

10, "Temperature is the source of heat and the manifestation of thermal energy [116], which flows from a higher to a lower level [117]." This is inaccurate and/or awkward. "Heat" and "thermal energy" are more or less synonyms. They both refer to energy that has not been given detailed accounting. E.g., to the translations, rotations, vibrations, etc. of molecules whose trajectories are not being observed or recorded directly, but rather are amalgamated non-specifically. "Temperature", at least in common thermodynamic usage, is a measure of the distribution of this thermal energy amongst the molecules or other particles that constitute the system in question. It is proportional to the width of the distribution, something like a standard deviation in statistics. But temperature is not the source of heat. A heat source is either a system of any kind that already contains this thermal energy at a higher temperature than some other system with which it is in contact, or a system containing energy in some higher form (e.g., mechanical energy, electrical energy, electromagnetic radiation,…) that is explicitly accounted for but that is being degraded into thermal energy.

We appreciate the suggestion, and accordingly, the necessary correction was made on p 10.

In an equilibrium system, temperature characterizes particles' kinetic energy. However, thermal energy measures the total kinetic energy of the particles [124] so that a heat source with thermal energy at a higher temperature flows toward a lower level at a lower temperature [125]. In psychology, emotional energy provides inspiration. For example, a message or post prompts an emotional distribution with a specific propagation probability between online social media users. Furthermore, the distribution of each sentiment's credibility, self-assertiveness, and other qualities form a decay function of sentiment contagion [126, 127].

11,  The authors go off on numerous tangents that are not clearly related to the main themes. This should be cut way back. The authors should avoid excursions and stay focused on developing the major topics.

We substantially streamlined the manuscript to channel its focus to the main subject. For example, we removed Table I and related text (p 10). We made a point to remove unrelated communication.

We want to thank the reviewer for his/her constructive criticism, which has improved our work quality significantly. We look forward to hearing from you regarding our submission and to respond to any further questions and comments you may have.

Sincerely,

Eva Deli

Author

Reviewer 2 Report

In their manuscript the Authors claim that they make case for thermodynamic explanation of emotions. They provide the neural background and give hints why such description could be promising. Numerous problems start to arise when they look for a thermodynamic framework which would fit their narrative. And the unfortunate result is massive hand-waving, poetic language, resorting to allegories and so on.

Thermodynamics is a well established science with numerous applications. First of all thermodynamic variables need to be identified, and a hint whether they are intensive of extensive, which are their conjugate pairs and so on. At some point entropy is called upon, but which one? The Authors do not distinguish thermodynamic entropy from the information entropy, a source of many pitfalls. As probing the brain is notoriously difficult, one is limited with experimental techniques at hand. fMRI, say, provides some information about slow-scale dynamics of a working brain, and such a measurement results in a time series for which an entropy related measure can be calculated, but that is not the entropy of a working brain, it is just a condense characterization of the measured time series, which is to a large degree a characterization of the space and time time resolution related problems of the measurement device, not the underlying processes it intends to measure. 

Brain is a system which is not in thermodynamic equilibrium, thus it spontaneously produces entropy at some rate. If you wish to understand brain or emotions in thermodynamic setting, that entropy production is relevant, not the entropy itself. Brain is an open system which is not homogeneous, description of such systems requires great care even for systems which are far simpler than the brain. 

This problem aside, what is offered in the manuscript? A set of trivial formulae copied from a first course textbook. If the Authors are serious with their intentions they need to develop a sound framework which identifies relations which involve temporal derivatives of thermodynamic potentials and relate them to entropy production, make an under-determined system of differential equations including thermodynamic variables and complete them with constitutive equations, your actual *model* of the brain, or emotions in this particular case. Because thermodynamic is ("just a") descriptive science, you have to provide the models, thermodynamics does not do that.

Table 1 is tragical. Why mental fermions are fermions, assuming they exist? Do they obey Fermi-Dirac statistics? What are mental bosons then?

The discussion of Carnot cycle in relation to neural processes unfortunately reflects deep misunderstanding of basic first year course thermodynamics.

Once these problems are solved, the intentions of the Authors may merit attention of the scientific community.

Author Response

Thank you for giving us the opportunity to submit a revised draft of our manuscript titled "HOW THE BRAIN BECOMES THE MIND: CAN THE BORN RULE AND THE CARNOT CYCLE EXPLAIN THE COGNITIVE PROCESS?" to the journal Entropy. We are grateful to the reviewer for the insightful comments. We have been able to incorporate changes to reflect most of the suggestions provided by the reviewers. We have used blue in our response below, but red within the manuscript to highlight the changes.

  1. Thermodynamics is a well established science with numerous applications. First of all thermodynamic variables need to be identified, and a hint whether they are intensive of extensive, which are their conjugate pairs and so on. At some point entropy is called upon, but which one? The Authors do not distinguish thermodynamic entropy from the information entropy, a source of many pitfalls. As probing the brain is notoriously difficult, one is limited with experimental techniques at hand. fMRI, say, provides some information about slow-scale dynamics of a working brain, and such a measurement results in a time series for which an entropy related measure can be calculated, but that is not the entropy of a working brain, it is just a condense characterization of the measured time series, which is to a large degree a characterization of the space and time time resolution related problems of the measurement device, not the underlying processes it intends to measure.

The nature of consciousness and emotions is crucial for psychology, psychiatry, and A.I. research. In our work we have been following other pioneering authors, who used thermodynamic principles in analyzing cognition (Collell and Fauquet, 2015; Deli et al., 2017; Yufik, 2013). Nevertheless, the field moves forward slowly due to the difficulties raised by the reviewer.

Classical thermodynamics provides a basis for a rigorous basis for the analysis of the behavior of a broad range of physical and chemical phenomena (Bratianu and Bejinaru, 2020). Although the derivation of neural behavior from fundamental physical laws is a difficult problem, we point to thermodynamics as a tangible basis for energy/information exchange of perception and the brain's computational role (Bratianu and Bejinaru, 2020). We introduce the Born rule to strengthen our argument (p 7.). The temporal derivation of the thermodynamic potentials is beyond the scope of the present work. 

We also elaborated on the information entropy as the number of neural states a given brain can access. Therefore, entropy became a powerful explanatory tool and a quantitative index of cognitive systems.

We characterized the ideal cognitive cycle on p 8. We applied the reversible Carnot cycle to characterize an ideal energy/information cycle between two cognitive states with vanishing net entropy production. The cycle's energy gain is conserved in the synaptic map, turning it an "invisible" factor in cognition structural relationships. The synaptic connections form an "independent medium," allowing the cycle to repeat with the same state variables of entropy, as a complexity-increasing harmonic motion. Therefore, entropy increases with time in a system moving to equilibrium (Cohen & Marron, 2020).

We hope that our investigation contributed to this exciting field.

  1. Brain is an open system which is not homogeneous, description of such systems requires great care even for systems which are far simpler than the brain. 
    We defined our statement regarding sensory perception more precisely. P 6.

The brain is considered an open system [77]. However, during energy-information exchange with the outside environment, the sensory stimulus goes through successive regulatory layers to the associate areas. Nevertheless, actions and responses restore the resting state while changing the neural landscape (the synaptic complexity) [78]. Therefore, sensory processing forms a closed cycle [26, 79, 58, 80, 81, 5]. From the physics perspective, a closed system exchanges only energy (as heat or work), not matter, with its surroundings, making the thermodynamic analysis of sensory input possible. Furthermore, Landauer's principle holds that irreversible manipulation of information increases entropy. Therefore, we can calculate the energy potential of the sensory information during the brain's intelligent computation.

We also inserted two new figures (Figures 3 and 4) to illustrate emotions' energy nature better, based on the new work by Gao et al. (2022).

In addition, we established the resting state, as the basis of the brain's thermodynamic cycle with the same state variables of entropy, p 5.

An intelligent answer to a stimulus requires an intuition of the physical world's organizational and operational principles: thoughts, intentions, and behavior form in symmetry with the environment [46]. Through constant alignment with the external world and action consequences, spontaneous resting activations are at the center of brain processing [60, 58, 59], which in turn formulate the sense of self [61, 62, 59] based on an internal model of the world.

  1. As requested, we removed Table 1 from the manuscript.

We want to thank the reviewer for his/her constructive criticism, which has improved our work quality significantly. We look forward to hearing from you regarding our submission and to respond to any further questions and comments you may have.

Sincerely,

Eva Deli

Author

Reviewer 3 Report

Brief summary:

The manuscript draws a parallel between the mind and thermodynamics. More specifically, the authors try to describe the thermodynamic origins of our emotions.

Overall Evaluation:

I have read several times this manuscript, and I now think that I have a good knowledge of it. The manuscript is well written (also see Line-by-line Comments), although for readers not well acquainted with thermodynamics, it is not easy to follow. The main problem that I have with the manuscript is that from my understanding no predictions are made. I think that if a new description/explanation of a phenomenon is presented it has to show what kind of predictions, and even better what kind of new predictions it can make. This would drastically increase the value of such new explanation. I therefore suggest some predictions should made (or put out more clearly) for the reader to better judge the heuristic power of such new description of the mind and of the emotions. If such predictions are made, I feel that the readers of the piece would not have any more the impression that it is just a big analogy. Personally, I believe that it is more than just an analogy and that it could be a valid tool. And I believe that a more convincing case could be made if more explicit predictions were put forward in the manuscript. I am asking for revision of the manuscript. I think that if the revision is done connivingly, it could even be achieved with a minor revision.

Line-by-line comment:

l.43

The authors write that the brain remain the least understood organ, maybe the authors could back it up with some references?

l.50

I beg to differ as the mind certainly starts to exist before birth (for example even just one minute before birth)

l.97

Could the authors elaborate as to why emotions are the ultimate source of actions?

l.142

The authors write:” They can trigger unpredictable and probabilistic behavior, forming psychology's reproducibility crisis.” Could the authors explain how this is linked with “psychology's reproducibility crisis”?

l.192

I am sure emotions are also linked to the frontal lobe, but it is not the principal region linked to emotions, for example the limbic system is more specifically linked to emotions.

Author Response

Thank you for the opportunity to submit a revised draft of the manuscript: " HOW THE BRAIN BECOMES THE MIND: CAN THE CARNOT CYCLE EXPLAIN THE EMERGENCE AND NATURE OF EMOTIONS?" We appreciate the time and expert attention to writing constructive and insightful feedback. The questions and concerns immensely helped us to think more systematically and comprehensively about these problems. We have closely followed the suggestions, insights, and criticism in the manuscript's revision.

We are thankful for the fair but thorough comments, which spurred significant improvements in the quality of the manuscript. As a result, the work became better focused and hopefully more relevant.

Below we addressed point-by-point the questions and comments in blue. All page numbers refer to the revised manuscript file. 

Brief summary:

The manuscript draws a parallel between the mind and thermodynamics. More specifically, the authors try to describe the thermodynamic origins of our emotions.

Overall Evaluation:

I have read several times this manuscript, and I now think that I have a good knowledge of it. The manuscript is well written (also see Line-by-line Comments), although for readers not well acquainted with thermodynamics, it is not easy to follow. The main problem that I have with the manuscript is that from my understanding no predictions are made. I think that if a new description/explanation of a phenomenon is presented it has to show what kind of predictions, and even better what kind of new predictions it can make. This would drastically increase the value of such new explanation. I therefore suggest some predictions should made (or put out more clearly) for the reader to better judge the heuristic power of such new description of the mind and of the emotions. If such predictions are made, I feel that the readers of the piece would not have any more the impression that it is just a big analogy. Personally, I believe that it is more than just an analogy and that it could be a valid tool. And I believe that a more convincing case could be made if more explicit predictions were put forward in the manuscript. I am asking for revision of the manuscript. I think that if the revision is done convincingly, it could even be achieved with a minor revision.

We added some instances on predictions:

P 12. The proposed hypothesis predicts a qualitative brain state difference between the decision preparation and post decision making. The first step’s reversible, chaotic and uncertain preparatory process is analog to the wave function of quantum mechanics. The first step is followed by certainty and unambiguity of the post decision. The thermodynamic cycle’s work potential (T2-T1) represents emotional intensity. Moreover, following dramatic life changes, interaction would exponentially decrease emotional intensity.  

P 11. However, the exothermic cycle's cognitive burden dissipates energy into the environment via destructive behavior. The free energy can drive internal hormone production with adverse mental and immune consequences. As we have shown, acceptance, which transforms the accumulated information into meaning and belief through Landauer's principle, can reverse the adverse consequences. Therefore, the analysis of possible changes in neurotransmitter action following cognitive loads might be fruitful. 

P 13. Our thermodynamic analysis of mental computation will permit the study of the brain's computational power.

Line-by-line comment:

l.43

The authors write that the brain remain the least understood organ, maybe the authors could back it up with some references?

We added the appropriate references.

R. Michel, M. Peters, D. Rahnev, C. Sergent and K. Liu, "An Informal Internet Survey on the Current State of Consciousness Science," Frontiers in Psychology, vol. 9, 2018.

M. Michel, D. Beck and N. Block, "Opportunities and challenges for a maturing science of consciousness," Nat Hum Behav, vol. 3, p. 104–107, 2019.

A. Nobre and F. van Ede, "Under the Mind's Hood: What We Have Learned by Watching the Brain at Work," The Journal of Neuroscience, vol. 40, pp. 89 - 100, 2020.

l.50

I beg to differ as the mind certainly starts to exist before birth (for example even just one minute before birth)

We corrected the mistaken statement: While the brain is within the skull, the mind exists in the present moment originating with the emergence of awareness.

l.97

Could the authors elaborate as to why emotions are the ultimate source of actions?

Response to stimulus depends on the appropriate reflection or mental model of the environment (Deli et al, 2021; Conant and Ashby, 1991). Our mental world is founded on beliefs i.e., the synaptic map, and changing it requires investing energy via mental work. Emotions, which indicate differences from expectation, motivate recovery of congruence, i.e., updating the mental model. 

l.142

The authors write:” They can trigger unpredictable and probabilistic behavior, forming psychology's reproducibility crisis.” Could the authors explain how this is linked with “psychology's reproducibility crisis”?

We inserted the sentence:

The stimulus triggers activations over the brain's internal oscillation pattern, which turns perception subjective. Therefore, the response is unpredictable and probabilistic, contributing to psychology's reproducibility crisis. Unlike a classical system, which remains unaffected by measurement, the brain's memory generation parallels a cognitive change [52] by connecting causality to motivation.

l.192

I am sure emotions are also linked to the frontal lobe, but it is not the principal region linked to emotions, for example the limbic system is more specifically linked to emotions.

We appreciate noting the mistake, and we made the following change:

Recent emotion studies indicate the cumulative effects of negative-emotion induction (i.e., fear, sadness, anger, and disgust) in the frontal lobe, which is entirely missing for neutral and happy sentiments [65].

We want to thank the referee again for the time, care, and give constructive suggestions on our work. We hope that the reviewer will be satisfied with our improved manuscript.

Round 2

Reviewer 1 Report

In response to my critiques, the authors have substantially revised their article resulting in a much improved manuscript.

Author Response

We want to thank the reviewer for the input on the manuscript: "HOW THE BRAIN BECOMES THE MIND: CAN THE CARNOT CYCLE EXPLAIN THE EMERGENCE AND NATURE OF EMOTIONS?" We are thankful to the referee for the fair but thorough comments, which resulted in a better-focused and more relevant manuscript.

Reviewer 2 Report

As I remarked in my review of the first version of the manuscript, and the claim holds for the revised one it is a collection of handwaving and wishful thinking written in a poetic language.  All the remarks written there still hold, I add some more.

The fermionic mind hypothesis (FMH) mentioned on p. 3 is utter nonsense. Luckily for the Authors, it has nothing to do with the Carnot-cycle, which is the central theme of the current version of the manuscript, so should be omitted.

To call something, a particle say, a fermion means that the particle follows Fermi Dirac statistics, obeys Pauli exclusion principle and is described with anticommuting variables, to say at least. None of these requirements is fulfilled. And then bosons are needed, otherwise no interaction is present.

Quantum mechanics is referred later on and Born's rule is mentioned. Again, it is completely unrelated to the Carnot cycle.

p. 6. 

Whole page is filled with nonsense, I note just a few examples.

Because entropy directs all changes toward equilibrium, it is connected to time-reversal asymmetry [67, 68, 69]. 
The entropy does not direct anything towards equilibrium, the entropy production does. 

brain entropy correlates with Shannon's entropy... 
How do you compute brain entropy to make such a statement? 

Therefore, the 'brain's intellectual computation is orthogonal vis-à-vis material systems. 
This statement is impossible to interprete in scientific terms. 

Landauer limit is totally misunderstood: 
...we can calculate the energy potential of the sensory information during the brain's intelligent computation...

The nervous system's superb adaption of the physical laws means that the mathematical formalism of quantum mechanics can successfully explain problems in psychology for decades [11] and our tendency for comparisons [12, 13, 14].    

The claim is very far from an established fact, irrespective of the citations.

subsection 2.2 is also a collection of misunderstandings, and that is the mildest way to put it. 

In biology the isothermal cycle has relevance, as the body temperature is more-less constant. 

To point a main weakness and a possibility to find a way out the mess, the Authors need to be crystal clear what they mean by temperature, no allegories of handwaving helps. 
Their proposed temperature is a measure of exactly what? In case of gases is related to mean kinetic energy, in case of other systems it is also related to some other known notions, so what is emotional temperature? What is the thermometer which measures it? Why Carnot-cycle? There are many other cycles which operate between two temperatures.  Why according to their wishes there are two temperatures? What are the heat reservoires, which are the heat exchange mechanisms? If it is Carnot-cycle is the heat exchange isothermal? Because it should be. And so much more ... 

Author Response

Thank you for the opportunity to submit a revised draft of the manuscript: " HOW THE BRAIN BECOMES THE MIND: CAN THE CARNOT CYCLE EXPLAIN THE EMERGENCE AND NATURE OF EMOTIONS?" We appreciate the time and expert attention to writing constructive and insightful feedback. The questions and concerns immensely helped us to think more systematically and comprehensively about these problems. We have closely followed the suggestions, insights, and criticism in the manuscript's revision.

We are thankful for the fair but thorough comments, which spurred significant improvements in the quality of the manuscript. As a result, the work became better focused and hopefully more relevant.

Below we addressed point-by-point the questions and comments in blue. All page numbers refer to the revised manuscript file. 

As I remarked in my review of the first version of the manuscript, and the claim holds for the revised one it is a collection of handwaving and wishful thinking written in a poetic language.  All the remarks written there still hold, I add some more.

The fermionic mind hypothesis (FMH) mentioned on p. 3 is utter nonsense. Luckily for the Authors, it has nothing to do with the Carnot-cycle, which is the central theme of the current version of the manuscript, so should be omitted.

We removed the expression, “fermionic mind hypothesis (FMH)” for the manuscript.

To call something, a particle say, a fermion means that the particle follows Fermi Dirac statistics, obeys Pauli exclusion principle and is described with anticommuting variables, to say at least. None of these requirements is fulfilled. And then bosons are needed, otherwise no interaction is present.

Quantum mechanics is referred later on and Born's rule is mentioned. Again, it is completely unrelated to the Carnot cycle.

First, Landauer's principle applies to irreversible processes. Irreversibility is the removal of generated heat and is a necessary requirement for the cycle to repeat. 

We analyze heat generation via thermodynamics. In the brain, the heat or information is transformed (i.e., erased) according to Landauer's principle, into synaptic complexity. 

We included the figure in the PDF file.

In the figure, the upward trending solid line represents the synaptic complexity. Therefore, synaptic complexity represents a stable, irreversible, and ultimately, classical condition.  

The ovals signify the thermodynamic cycle, reversible, probabilistic, and uncertain, therefore, quantum-like behavior. 

The above argument shows that for the thermodynamic cycle to repeat exactly it is necessary to erase the accumulated information of the cycle. Landauer’s principle shows that erasing information has energetic consequences, thus, leading to the accumulation of synaptic complexity. 

Synaptic complexity, the basis of beliefs and personality

The brain’s thermodynamic cycle, triggered by sensory experience, memories, thoughts

Predictable

Probabilistic

Not reversible without investing mental effort

Reversible

Stable

Transient

Forming a field (classical behavior)

Quantum-like behavior

Therefore, we did view the thermodynamic cycle as a quantum-like system.

  1. 6. 

Whole page is filled with nonsense, I note just a few examples.

Because entropy directs all changes toward equilibrium, it is connected to time-reversal asymmetry [67, 68, 69]. 
The entropy does not direct anything towards equilibrium, the entropy production does. 

On p.6, we made the requested change: Because entropy production directs all changes toward equilibrium, it is connected to time-reversal asymmetry [67, 68, 69]. 

brain entropy correlates with Shannon's entropy... 
How do you compute brain entropy to make such a statement? 

We made the appropriate change in the manuscript. In information theory, Shannon's entropy, the entropy of a random variable is the average level of "information", "surprise", or "uncertainty" inherent to the variable's possible outcomes. Specifically, the original Shannon entropy was proposed for the case of discrete random variables, whereas the computation of brain entropy, such as Differential Entropy (DE), Permutation Entropy (PE), sample entropy (SE), and (Multiscale Entropy) MSE are based on estimates of the entropy of a continuous-time series through its discretization. 

Therefore, the 'brain's intellectual computation is orthogonal vis-à-vis material systems. 
This statement is impossible to interprete in scientific terms. 

We removed the criticized statement: the 'brain's intellectual computation is orthogonal vis-à-vis material systems. 

Landauer limit is totally misunderstood: 
...we can calculate the energy potential of the sensory information during the brain's intelligent computation...

On p 7. Landauer's principle pertains to the energy consumption of computation. Specifically, the principle holds that the irreversible manipulation of information increases entropy; therefore, it can be applied to cognitive computation. Furthermore, Landauer's principle shows that information erasure, such as learning, meaning generation, acceptance, and catharsis, requires entropy. Therefore, we can calculate the energy requirement of information erasure during the brain's intelligent computation.

The nervous system's superb adaption of the physical laws means that the mathematical formalism of quantum mechanics can successfully explain problems in psychology for decades [11] and our tendency for comparisons [12, 13, 14].    

The claim is very far from an established fact, irrespective of the citations.

Quantum cognition, which explains the cognitive process by using the mathematical formalism of quantum mechanics, has a decades-long history. It provides the best explanation for many phenomena in psychology and social sciences to date. We believe that ever since the first scientific endeavor of mankind in antiquity, disagreements were a necessary part and perhaps the best drivers of progress. Nevertheless, we will be pleased to use an alternative expression proposed by the referee that provides a satisfactory explanatory power.

subsection 2.2 is also a collection of misunderstandings, and that is the mildest way to put it. 

We strengthened our arguments in subsection 2.2:

When the cognitive system encounters supportive evidence that it is grounded on perennial and appropriate foundations, the experience of pleasure encourages more allocation of attentional resources to the subject (Schoeller & Perlovsky, 2016). The strength of emotion depends on meaning-making and psychologically relevant contents (Schoeller and Perlovsky, 2015).

Thus, access to air, water, food, and mating improves survival or life expectancy. Evolution became an arms race to improve access to time, which translates into information. Emotions measure access to time by continuously changing between urgency and relaxation.

In biology the isothermal cycle has relevance, as the body temperature is more-less constant. 

To point a main weakness and a possibility to find a way out the mess, the Authors need to be crystal clear what they mean by temperature, no allegories of handwaving helps. 
Their proposed temperature is a measure of exactly what? In case of gases is related to mean kinetic energy, in case of other systems it is also related to some other known notions, so what is emotional temperature? What is the thermometer which measures it? Why Carnot-cycle? There are many other cycles which operate between two temperatures.  Why according to their wishes there are two temperatures? What are the heat reservoires, which are the heat exchange mechanisms? If it is Carnot-cycle is the heat exchange isothermal? Because it should be. And so much more ...  

Social temperature represents nervousness, such as speculative risk-taking (Krause & Bornholdt, 2012). Anger and excitement, and accompanying increased breath and heart rate raise body temperature; major emotional events often trigger hot or cold chills. In addition, fever, and psychogenic shivers have metabolic costs. The above psychological processes exert a measurable metabolic burden on the body (Schoeller and Perlovsky, 2016). Although anger, shame, and other adverse emotions represent higher frequencies and greater metabolic costs primarily affect the brain, and produce physical outbursts at their extreme, such as shouting, pacing, and aggression, with significant energy requirements.

Because particle motion is the function of temperature, social competition parallels social temperature. It is a measure of irritability and excitement, accompanied by increased breath and heart rate, triggering risky behavior [91]. It often can cause measurable changes in body temperature; major emotional events often trigger hot or cold chills. Like a fever, psychogenic shivers and other nervous states exert a measurable metabolic burden on the body [92]. Although anger, shame, and other adverse emotions representing higher frequencies increase the brain’s metabolic costs, at their extreme, they produce physical work, such as shouting, pacing, and aggression. Therefore, agitation and other nervous states, which instigate struggle between community members, represent high social temperatures.

We appreciate the time and critical attention of the reviewer, which forced us to think about the work more methodically and systematically than before. We worked hard in making the requested changes out of respect for the referee’s constructive and expert criticism and our conviction that our work examines important questions, relating to the nature of intelligent computation.   

Reviewer 3 Report

The authors have answered my questions and dealt with my comments well. I have nothing to add and accept the manuscript.

Round 3

Reviewer 2 Report

Intentionally left blank.